# Regularizing Neural Networks
# with Meta-Learning Generative Models

**Shin'ya Yamaguchi**[†,‡]* **Daiki Chijiwa**[†] **Sekitoshi Kanai**[†]
**Atsutoshi Kumagai**[†] **Hisashi Kashima**[‡]
[†]NTT  [‡]Kyoto University

## Abstract

This paper investigates methods for improving generative data augmentation for deep learning. Generative data augmentation leverages the synthetic samples produced by generative models as an additional dataset for classification with small dataset settings. A key challenge of generative data augmentation is that the synthetic data contain uninformative samples that degrade accuracy. This is because the synthetic samples do not perfectly represent class categories in real data and uniform sampling does not necessarily provide useful samples for tasks. In this paper, we present a novel strategy for generative data augmentation called *meta generative regularization* (MGR). To avoid the degradation of generative data augmentation, MGR utilizes synthetic samples in the regularization term for feature extractors instead of in the loss function, e.g., cross-entropy. These synthetic samples are dynamically determined to minimize the validation losses through meta-learning. We observed that MGR can avoid the performance degradation of naïve generative data augmentation and boost the baselines. Experiments on six datasets showed that MGR is effective particularly when datasets are smaller and stably outperforms baselines.

## 1 Introduction

While deep neural networks achieved impressive performance on various machine learning tasks, training them still requires a large amount of labeled training data in supervised learning. The labeled datasets are expensive when a few experts can annotate the data, e.g., medical imaging. In such scenarios, *generative data augmentation* is a promising option for improving the performance of models. Generative data augmentation basically adds pairs of synthetic samples from conditional generative models and their target labels into real training datasets. The expectations of generative data augmentation are that the synthetic samples interpolate missing data points and perform as oversampling for classes with less real training samples [1]. This simple method can improve the performance of several tasks with less diversity of inputs such as medical imaging tasks [2, 3, 4, 5].

However, in general cases that require the recognition of more diverse inputs (e.g., CIFAR datasets [6]), generative data augmentation degrades rather than improves the test accuracy [7]. Previous studies have indicated that this can be caused by the low quality of synthetic samples in terms of the diversity and fidelity [7, 8]. If this hypothesis is correct, we can expect high-quality generative models (e.g., StyleGAN2-ADA [9]) to resolve the problem; existing generative data augmentation methods adopt earlier generative models e.g., ACGAN [10] and SNGAN [11]. Contrary to the expectation, this is not the case. We observed that generative data augmentation fails to improve models even when using a high-quality StyleGAN2-ADA (Figure 1). Although the samples partially appear to be real to humans, they are not yet sufficient to train classifiers in

---

*Corresponding author. Email: `shinya.yamaguchi@ntt.com`

37th Conference on Neural Information Processing Systems (NeurIPS 2023).

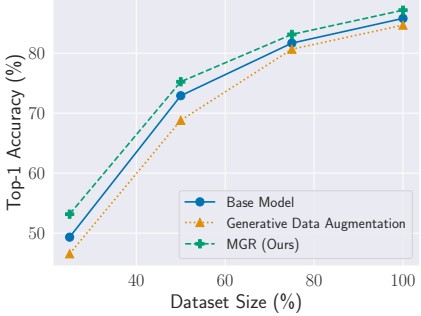

Figure 1: Accuracy gain using meta generative regularization on Cars [12] with ResNet-18 classifier [13] and StyleGAN2-ADA [9] (FID: 9.5)

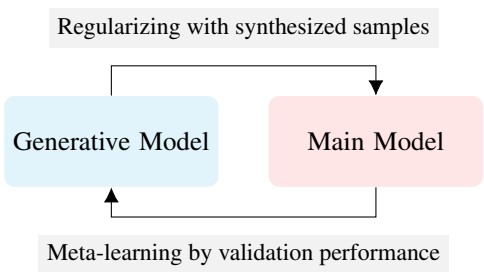

Figure 2: Meta generative regularization

existing generative data augmentation methods. This paper investigates methodologies for effectively extracting useful information from generative models to improve model performance.

We address this problem based on the following hypotheses. First, *synthetic samples are actually informative but do not perfectly represent class categories in real data*. This is based on a finding by Brock et al. [14] called "class leakage," where a class conditional synthetic sample contains attributes of other classes. For example, they observed failure samples including an image of "tennis ball" containing attributes of "dogs" (Figure 4(d) of [14]). These class leaked samples do not perfectly represent the class categories in the real dataset. If we use such class leaked samples for updating classifiers, the samples can distort the decision boundaries, as shown in Figure 3. Second, *regardless of the quality, the generative models originally contain uninformative samples to solve the tasks*. This is simply because the generative models are not explicitly optimized to generate informative samples for learning the conditional distribution $p(y|x)$; they are optimized only for learning the data distribution $p(x)$. Further, the generative models often fail to capture the entire data distribution precisely due to their focus on the high-density regions. These characteristics of generative models might disturb the synthesis of effective samples for generative data augmentation. To maximize the gain from synthetic samples, we should select appropriate samples for training tasks.

In this paper, we present a novel regularization method called *meta generative regularization* (MGR). Based on the above hypotheses, MGR is composed of two techniques for improving generative data augmentation: *pseudo consistency regularization* (PCR) and *meta pseudo sampling* (MPS). PCR is a regularization term using synthetic samples in training objectives for classifiers. Instead of supervised learning with negative log-likelihood $-\log p(y|x)$, i.e., cross-entropy, on synthetic samples, we regularize the feature extractor to avoid the distortions on decision boundaries. That is, PCR leverages synthetic samples only for learning feature spaces. PCR penalizes the feature extractors by minimizing the gap between variations of a synthetic sample, which is inspired by consistency regularization in semi-supervised learning [15, 16, 17]. MPS corresponds to the second hypothesis and its objective is to select useful samples for training tasks by dynamically searching optimal latent vectors of the generative models. Therefore, we formalize MPS as a bilevel optimization framework of a classifier and a finder that is a neural network for searching latent vectors. Specifically, this framework updates the finder through meta-learning to reduce the validation loss and then updates the classifier to reduce the PCR loss (Figure 2). By combining PCR and MPS, we can improve the performance even when the existing generative data augmentation degrades the performance (Figure 1).

We conducted experiments with multiple vision datasets and observed that MGR can stably improve baselines on various settings by up to 7 percentage points of test accuracy. Further, through the visualization studies, we confirmed that MGR utilizes the information in synthetic samples to learn feature representations through PCR and obtain meaningful samples through MPS.

## 2 Preliminary

### 2.1 Problem Setting

We consider a classification problem in which we train a neural network model $f_\theta : \mathcal{X} \to \mathcal{Y}$ on a labeled dataset $\mathcal{D} = \{(x^i, y^i) \in \mathcal{X} \times \mathcal{Y}\}_{i=1}^N$, where $\mathcal{X}$ and $\mathcal{Y}$ are the input and output label spaces,

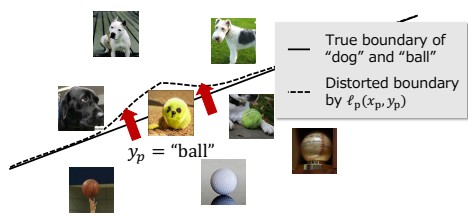

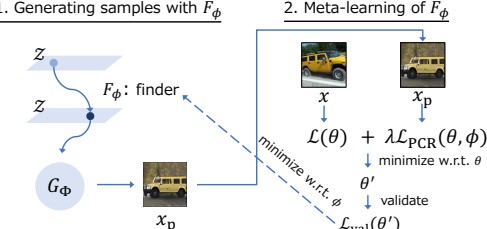

Figure 3: Distortion of decision boundary caused by generative data augmentation with conditional synthetic samples leaking another class attribute. If the synthetic sample $x_{\mathrm{p}}$ is "tennis ball dog" (reprinted from [18]) with its conditional label $y_{\mathrm{p}} = $ "ball", the supervised learner of a task head $h_\omega$ distorts the decision boundary of "dog" and "ball" to classify $x_{\mathrm{p}}$ as "ball".

Figure 4: Meta pseudo sampling framework. We meta-optimize the finder $F_\phi$ to generate a useful latent vector $F_\phi(z)$ for training a model parameter $\theta$ through minimizing the validation loss with the once updated parameter $\theta'$ by a real sample $x$ and a synthetic sample $x_{\mathrm{p}} = G_\Phi(F_\phi(z))$.

respectively. Here, we can use a generative model $G_\Phi : \mathcal{Z} \times \mathcal{Y} \to \mathcal{X}$, which is trained on $\mathcal{D}$. We assume that $G_\Phi$ generates samples from a latent vector $z \in \mathcal{Z}$ and conditions on samples with a categorical label $y \in \mathcal{Y}$, where $z$ is sampled from a standard Gaussian distribution $p(z) = \mathcal{N}(0, I)$ and $y$ is uniformly sampled from $\mathcal{Y}^2$. We refer to the classification task on $\mathcal{D}$ as the main task, and $f_\theta$ as the main model. $f_\theta$ is defined by a composition of a feature extractor $g_\psi$ and a classifier $h_\omega$, i.e., $f_\theta = h_\omega \circ g_\psi$ and $\theta = [\psi, \omega]$. To validate $f_\theta$, we can use a small validation dataset $\mathcal{D}_{\mathrm{val}} = \{(x_{\mathrm{val}}^i, y_{\mathrm{val}}^i) \in \mathcal{X} \times \mathcal{Y}\}_{i=1}^{N_{\mathrm{val}}}$, which has no intersection with $\mathcal{D}$ (i.e., $\mathcal{D} \cap \mathcal{D}_{\mathrm{val}} = \emptyset$).

## 2.2 Generative Data Augmentation

A typical generative data augmentation trains a main model $f_\theta$ with both real data and synthetic data from the generative models [19]. We first generate synthetic samples to be utilized as additional training data for the main task. Most previous studies on generative data augmentation [19, 20, 8] adopt conditional generative models for $G_\Phi$, and generate a pseudo dataset $\mathcal{D}_{\mathrm{p}}$ as

$$\mathcal{D}_{\mathrm{p}} = \{(x_{\mathrm{p}}^i, y_{\mathrm{p}}^i) \mid x_{\mathrm{p}}^i = G_\Phi(z^i, y_{\mathrm{p}}^i)\}_{i=1}^{N_{\mathrm{p}}}, \tag{1}$$

where $z^i$ is sampled from a prior distribution $p(z)$, and $y_{\mathrm{p}}^i$ is uniformly sampled from $\mathcal{Y}$. Subsequently, $f_\theta$ is trained on both of $\mathcal{D}$ and $\mathcal{D}_{\mathrm{p}}$ using the following objective function.

$$\min_\theta \quad \mathcal{L}(\theta) + \lambda \mathcal{L}_p(\theta), \tag{2}$$

$$\mathcal{L}(\theta) = \mathbb{E}_{(x,y)\in\mathcal{D}}\ell(f_\theta(x), y), \tag{3}$$

$$\mathcal{L}_{\mathrm{p}}(\theta) = \mathbb{E}_{(x_{\mathrm{p}},y_{\mathrm{p}})\in\mathcal{D}_{\mathrm{p}}}\ell_{\mathrm{p}}(f_\theta(x_{\mathrm{p}}), y_{\mathrm{p}}), \tag{4}$$

where $\ell$ is a loss function of the main task (e.g., cross-entropy), $\ell_{\mathrm{p}}$ is a loss function for the synthetic samples, and $\lambda$ is a hyperparameter for balancing $\mathcal{L}$ and $\mathcal{L}_{\mathrm{p}}$. In previous works, $\ell_{\mathrm{p}}$ is often set the same as $\ell$. Although optimizing Eq. (2) with respect to $\theta$ is expected to boost the test performance by interpolating or oversampling conditional samples [1], its naïve application degrades the performance of $f_\theta$ on general settings [7]. In this paper, we explore methods to resolve the degradation of generative data augmentation and maximize the performance gain from $\mathcal{D}_{\mathrm{p}}$.

## 3 Proposed Method

In this section, we describe our proposed method, MGR. The training using MGR is formalized as alternating optimization of a main model and finder network for searching latent vectors of $G_\Phi$, as shown in Figure 2. To maximize the gain from synthetic samples, MGR regularizes a feature extractor $g_\psi$ of $f_\theta$ using PCR by effectively sampling useful samples for the generalization from $G_\Phi$ using MPS. We show the overall algorithm of MGR in Appendix A.

---

$^2$ Although we basically use conditional $G_\Phi$ for comparing our method and generative data augmentation, our method can be used with unconditional $G_\Phi$ (see Sec. 4.6).

## 3.1 Pseudo Consistency Regularization

As discussed in Secion 1, we hypothesize that the synthetic samples do not perfectly represent class categories, and training classifiers using them can distort the decision boundaries. This is because $y_p$ is not reliable due to $\mathcal{D}_p$ can contain class leaked samples [14]. To avoid the degradation caused by the distortion, we propose utilizing $x_p$ to regularize only the feature extractor $g_\psi$ of $f_\theta$ by discarding a conditional label $y_p$. For the regularization, we borrow the concept of consistency regularization, which was originally proposed for semi-supervised learning (SSL) [15, 16, 17]. These SSL methods were designed to minimize the dissimilarity between the two logits (i.e., the output of $h_\omega$) of strongly and weakly transformed unlabeled samples to obtain robust representations. By following this concept, the PCR loss is formalized as

$$\ell_{\text{PCR}}(x_p; \psi) = \|g_\psi(T(x_p)) - g_\psi(x_p)\|_2^2, \tag{5}$$

where $T$ is a strong transformation such as RandAugment [21], which is similar to the one used in UDA [16] for SSL. The difference between PCR and UDA is that PCR penalizes only $g_\psi$, whereas UDA trains the entire $f_\theta = h_\omega \circ g_\psi$. $\ell_{\text{PCR}}$ can be expected to help $g_\psi$ learns features of inter-cluster interpolated by $x_p$ without distorting the decision boundaries. Using $\ell_{\text{PCR}}$, we rewrite Eq. (4) as

$$\mathcal{L}_{\text{PCR}}(\theta) = \mathbb{E}_{x_p \in \mathcal{D}_p} \ell_{\text{PCR}}(x_p; \psi). \tag{6}$$

## 3.2 Meta Pseudo Sampling

Most generative data augmentation methods generate a synthetic sample $x_p$ from $G_\Phi$ with a randomly sampled latent vector $z$. This is not the best option as $G_\Phi$ is not optimized for generating useful samples to train $f_\theta$ in predicting the conditional distribution $p(y|x)$; it is originally optimized to replicate $p(x|y)$ or $p(x)$. The main concept of MPS is to directly determine useful samples for training $f_\theta$ by optimizing an additional neural network called a *finder* $F_\phi : \mathcal{Z} \to \mathcal{Z}$. $F_\phi$ takes a latent vector $z \sim p(z)$ as input and outputs a vector of the same dimension as $z$. By using $F_\phi$, we generate a synthetic sample as $x_p = G_\Phi(F_\phi(z), y)$. The role of $F_\phi$ is to find the optimal latent vectors that improve the training of $f_\theta$ through $x_p$. Although we can consider optimizing $G_\Phi$ instead of $F_\phi$, we optimize $F_\phi$ because the previous work showed that transforming latent variables according to loss functions efficiently reaches the desired outputs [22] and we observed that optimizing $G_\Phi$ is unstable and causes the performance degradations of $f_\theta$ (Section 4.5).

The useful samples for generalization should reduce the validation loss of $f_\theta$ by using them for optimization. Based on this simple suggestion, we formalize the following bilevel optimization problem for $F_\phi$.

$$\min_\phi \mathcal{L}_{\text{val}}(\theta^*) = \mathbb{E}_{(x_{\text{val}}, y_{\text{val}}) \in \mathcal{D}_{\text{val}}} \ell(f_{\theta^*}(x_{\text{val}}), y_{\text{val}})$$
$$\text{subject to} \qquad \theta^* = \operatorname*{argmin}_\theta \mathcal{L}(\theta) + \lambda \mathcal{L}_{\text{PCR}}(\theta, \phi). \tag{7}$$

Note that the finder parameter $\phi$ is added to the arguments of $\mathcal{L}_{\text{PCR}}$. We can optimize $F_\phi$ with a standard gradient descent algorithm because $F_\phi$, $G_\Phi$, and $f_\theta$ are all composed of differentiable functions as well as existing meta-learning methods such as MAML [23] and DARTS [24]. We approximate $\theta^*$ because the exact computation of the gradient $\nabla_\phi \mathcal{L}_{\text{val}}(\theta^*)$ is expensive [23, 24]:

$$\theta^* \approx \theta' = \theta - \eta \nabla_\theta(\mathcal{L}(\theta) + \lambda \mathcal{L}_{\text{PCR}}(\theta, \phi)), \tag{8}$$

where $\eta$ is an inner step size. Thus, we update $F_\phi$ by using $\theta'$ that is updated for a single step and then alternately update $\theta$ by applying Eq. (6) with $x_p$ generated from the updated $F_\phi$. The overall optimization flow is shown in Figure 4.

**Approximating gradients with respect to $\phi$.** Computing $\nabla_\phi \mathcal{L}_{\text{val}}(\theta')$ requires the product computation including second-order gradients: $\nabla^2_{\phi,\theta} \mathcal{L}_{\text{PCR}}(\theta, \phi) \nabla_{\theta'} \mathcal{L}_{\text{val}}(\theta')$. This causes a computation complexity of $\mathcal{O}((|\Phi| + |\phi|)|\theta|)$. To avoid this computation, we approximate the term using the finite difference method [24] as

$$\nabla^2_{\phi,\theta} \mathcal{L}_{\text{PCR}}(\theta, \phi) \nabla_{\theta'} \mathcal{L}_{\text{val}}(\theta') \approx \frac{\nabla_\phi \mathcal{L}_{\text{PCR}}(\theta^+, \phi) - \nabla_\phi \mathcal{L}_{\text{PCR}}(\theta^-, \phi)}{2\varepsilon}, \tag{9}$$

where $\theta^\pm$ is $\theta$ updated by $\theta \pm \varepsilon \eta \nabla_\theta \mathcal{L}_{\text{val}}(\theta)$. $\varepsilon$ is defined by $\frac{\text{const.}}{\|\nabla_\theta \mathcal{L}_{\text{val}}(\theta)\|_2}$. We used 0.01 of the constant for $\varepsilon$ based on [24]. This approximation reduces the computation complexity to $\mathcal{O}(|\Phi| + |\phi| + |\theta|)$. We confirm the speedup by Eq. (9) in Appendix B.1.

**Techniques for improving finder**   We introduce two techniques for improving the training of $F_\phi$ in terms of the architectures and a penalty term for the outputs.

While arbitrary neural architectures can be used for the implementation of $F_\phi$, we observed that the following residual architecture produces better results.

$$F_\phi(z) := z + \tanh(\text{MLP}_\phi(z)), \tag{10}$$

where MLP is multi layer perceptron.

To ensure that $F_\phi(z)$ does not diverge too far from the distribution $p(z)$, we add a Kullback–Leibler divergence term $D_{\text{KL}}(p_\phi(z)\|p(z))$, where $p_\phi(z)$ is the distribution of $F_\phi(z)$ into Eq. (7). When $p(z)$ follows the standard Gaussian $\mathcal{N}(0, I)$, $D_{\text{KL}}(p_\phi(z)\|p(z))$ can be computed by

$$D_{\text{KL}}(p_\phi(z)\|p(z)) = -\frac{1}{2}(1 + \log \sigma_\phi - \mu_\phi^2 - \sigma_\phi), \tag{11}$$

where $\mu_\phi$ and $\sigma_\phi$ is the mean and variance of $\{F_\phi(z^i)\}_{i=1}^{N_{\text{P}}}$. In Appendix B.2, we discuss the effects of design choice based on the ablation study.

## 4   Experiments

In this section, we evaluate our MGR (the combination of PCR and MPS with the experiments on multiple image classification datasets. We mainly aim to answer three research questions with the experiments: (1) Can PCR avoid the negative effects of $x_{\text{p}}$ in existing methods and improve the performance of $f_\theta$? (2) Can MPS find better samples for training than those by uniform sampling? (3) How practical is the performance of MGR? We compare MGR with baselines including conventional generative data augmentation and its variants in terms of test performance (Sec. 4.2 and 4.3). Furthermore, we conduct a comprehensive analysis of PCR and MPS such as the visualization of trained feature spaces (Sec. 4.4), quantitative/qualitative evaluations of the synthetic samples (Sec. 4.5), performance studies when changing generative models (Sec. 4.6), and comparison to data augmentation methods such as TrivialAugment [25] (Sec. 4.7).

### 4.1   Settings

**Baselines.**   We compare our method with the following baselines. **Base Model**: training $f_\theta$ with only $\mathcal{D}$. **Generative Data Augmentation (GDA)**: training $f_\theta$ with $\mathcal{D}$ and $G_\Phi$ using Eq. (2). **GDA+MH**: training $f_\theta$ with $\mathcal{D}$ and $G_\Phi$ by decoupling the heads into $h_\omega$ for $\mathcal{D}$ and $h_{\omega_{\text{p}}}$ for $\mathcal{D}_{\text{p}}$. MH denotes multi-head. This is a naïve approach to avoid the negative effect of $x_{\text{p}}$ on $h_\omega$ by not passing $x_{\text{p}}$ through $h_\omega$. GDA+MH optimizes the parameters as $\underset{\theta}{\arg\min}\,\mathcal{L}(f_\theta(x), y) + \lambda\mathcal{L}(h_{\omega_{\text{p}}}(g_\psi(x_{\text{p}})), y_{\text{p}})$.

**GDA+SSL**: training $f_\theta$ with $\mathcal{D}$ and $G_\Phi$ by applying an SSL loss for $\mathcal{D}_{\text{p}}$ that utilizes the output of $h_\omega$ unlike PCR. This method was originally proposed by Yamaguchi et al. [26] for transfer learning, but we note that $G_\Phi$ was trained on the main task dataset $\mathcal{D}$ in contrast to the original paper. By following [26], we used UDA [16] as the SSL loss and the same strong transformation $T$ as of MGR for the consistency regularization.

**Datasets.**   We used six image datasets for classification tasks in various domains: Cars [12], Aircraft [27], Birds [28], DTD [29], Flowers [30], and Pets [31]. Furthermore, to evaluate smaller dataset cases, we used subsets of Cars that were reduced by $\{10, 25, 50, 75\}\%$ in volume; we reduced them by random sampling on a fixed random seed. We randomly split a dataset into $9 : 1$ and used the former as $\mathcal{D}$ and the latter as $\mathcal{D}_{\text{val}}$.

**Architectures.**   We used ResNet-18 [13] as $f_\theta$ and generators of conditional StyleGAN2-ADA for $256 \times 256$ images [18] as $G_\Phi$. $F_\phi$ was composed of a three-layer perceptron with a leaky-ReLU activation function. We used the ImageNet pre-trained weights of ResNet-18 distributed by PyTorch.[3] For StyleGAN2-ADA, we did not use pre-trained weights. We trained $G_\Phi$ on each $\mathcal{D}$ from scratch according to the default setting of the implementation of StyleGAN2-ADA.[4] Note that we used the same $G_\Phi$ in the baselines and our proposed method.

---

[3] https://github.com/pytorch/vision   [4] https://github.com/NVlabs/stylegan2-ada

Table 1: Top-1 accuracy (%) of ResNet18. Underlined scores outperform that of Base Model, and **Bolded scores** are the best among the methods.

(a) Multiple Datasets

| Method / Dataset | Cars | Aircraft | Birds | DTD | Flower | Pets |
|---|---|---|---|---|---|---|
| Base Model | $85.80^{\pm.10}$ | $62.61^{\pm.79}$ | $72.24^{\pm.32}$ | $68.16^{\pm.35}$ | $94.18^{\pm.08}$ | $87.21^{\pm.13}$ |
| GDA | $84.50^{\pm.25}$ | $61.29^{\pm.05}$ | $67.55^{\pm.11}$ | $67.68^{\pm.37}$ | $93.46^{\pm.15}$ | $\underline{87.37}^{\pm.21}$ |
| GDA+MH | $85.77^{\pm.10}$ | $\underline{63.16}^{\pm.15}$ | $71.95^{\pm.09}$ | $\underline{68.49}^{\pm.21}$ | $93.83^{\pm.11}$ | $\underline{87.77}^{\pm.10}$ |
| GDA+SSL | $85.75^{\pm.21}$ | $62.87^{\pm.54}$ | $71.28^{\pm.42}$ | $\underline{68.49}^{\pm.59}$ | $93.66^{\pm.08}$ | $\underline{87.75}^{\pm.21}$ |
| GDA+MPS | $85.28^{\pm.05}$ | $62.11^{\pm.34}$ | $72.25^{\pm.13}$ | $\underline{68.87}^{\pm.22}$ | $\underline{94.39}^{\pm.14}$ | $\underline{87.92}^{\pm.11}$ |
| PCR | $\underline{86.36}^{\pm.08}$ | $\underline{64.43}^{\pm.21}$ | $\underline{73.69}^{\pm.10}$ | $\underline{69.17}^{\pm.18}$ | $\underline{94.66}^{\pm.22}$ | $\underline{88.59}^{\pm.44}$ |
| MGR (PCR+MPS) | $\mathbf{87.22}^{\pm.15}$ | $\mathbf{65.11}^{\pm.57}$ | $\mathbf{74.24}^{\pm.34}$ | $\mathbf{69.53}^{\pm.28}$ | $\mathbf{95.42}^{\pm.20}$ | $\mathbf{88.98}^{\pm.01}$ |

(b) Small Datasets (Cars)

| Method / Dataset size | 10% | 25% | 50% | 75% |
|---|---|---|---|---|
| Base Model | $20.11^{\pm.03}$ | $49.33^{\pm.54}$ | $72.91^{\pm.38}$ | $81.68^{\pm.18}$ |
| GDA | $18.91^{\pm.54}$ | $46.56^{\pm.07}$ | $68.81^{\pm.52}$ | $80.67^{\pm.10}$ |
| GDA+MH | $18.40^{\pm.93}$ | $46.63^{\pm.32}$ | $71.51^{\pm.10}$ | $81.25^{\pm.32}$ |
| GDA+SSL | $19.71^{\pm.41}$ | $47.80^{\pm.30}$ | $71.99^{\pm.40}$ | $81.55^{\pm.38}$ |
| MGR | $\mathbf{23.49}^{\pm.53}$ | $\mathbf{53.16}^{\pm.32}$ | $\mathbf{75.24}^{\pm.21}$ | $\mathbf{83.13}^{\pm.27}$ |

**Training.** We trained $f_\theta$ by the Nesterov momentum SGD for 200 epochs with a momentum of 0.9, and an initial learning rate of 0.01; we decayed the learning rate by 0.1 at 60, 120, and 160 epochs. We trained $F_\phi$ by the Adam optimizer for 200 epochs with a learning rate of $1.0 \times 10^{-4}$. We used mini-batch sizes of 64 for $\mathcal{D}$ and 64 for $\mathcal{D}_\mathrm{p}$. The input samples were resized into a resolution of $224 \times 224$; $x_\mathrm{p}$ was resized by differentiable transformations. For synthetic samples from $G_\Phi$ in PCR and GDA+SSL, the strong transformation $T$ was RandAugment [21] by following [16], and it was implemented with differentiable transformations provided in Kornia [32]. We determined the hyperparameter $\lambda$ by grid search among $[0.1, 1.0]$ with a step size of $0.1$ for each method by $\mathcal{D}_\mathrm{val}$. To avoid overfitting, we set the hyperparameters of MGR that are searched with only applying PCR i.e., we did not use meta-learning to choose them. We used $\lambda_\mathrm{KL}$ of $0.01$. We selected the final model by checking the validation accuracy for each epoch. We ran the experiments three times on a 24-core Intel Xeon CPU with an NVIDIA A100 GPU with 40GB VRAM and recorded average test accuracies with standard deviations evaluated on the final models.

## 4.2 Evaluation on Multiple Datasets

We confirm the efficacy of MGR across multiple datasets. Table 1a shows the top-1 accuracy scores of each method. As reported in [7], GDA degraded the base model on many datasets; it slightly improved the base model on only one dataset. GDA+MH, which had decoupled classifier heads for GDA, exhibited a similar trend to GDA. This indicates that simply decoupling the classifier heads is not a solution to the performance degradation caused by synthetic samples. In contrast, our MGR stably and significantly outperformed the baselines and achieved the best results. The ablation of MGR discarding PCR or MPS is listed in Table 1a. We confirm that both PCR and GDA+MPS improve GDA. While GDA+SSL underperforms the base models, PCR outperforms the base models. This indicates that using unsupervised loss alone is not sufficient to eliminate the negative effects of the synthetic samples and that discarding the classifier $h_\omega$ from the regularization is important to obtain the positive effect. MPS yields only a small performance gain when combined with GDA, but it significantly improves its performance when combined with PCR i.e., MGR. This suggests that there is no room for performance improvements in GDA, and MPS can maximize the potential benefits of PCR.

## 4.3 Evaluation on Small Datasets

A small dataset setting is one of the main motivations for utilizing generative data augmentation. We evaluate the effectiveness of MGR on smaller datasets. Table 1b shows the performance when reducing the Cars dataset into a volume of $\{10, 25, 50, 75\}\%$. Note that we trained $G_\Phi$ on each reduced dataset, not on $100\%$ of Cars. In contrast to the cases of the full dataset (Table 1a), no baseline methods outperformed the base model in this setting. This is because $G_\Phi$ trained on the small datasets generates low-quality samples with less reliability on the conditional label $y_\mathrm{p}$ that are not appropriate in supervised learning. On the other hand, MGR improved the baselines in large margins. This indicates that, even when the synthetic samples are not sufficient to represent the class categories, our MGR can maximize the information obtained from the samples by utilizing them to regularize feature extractors and dynamically finding useful samples.

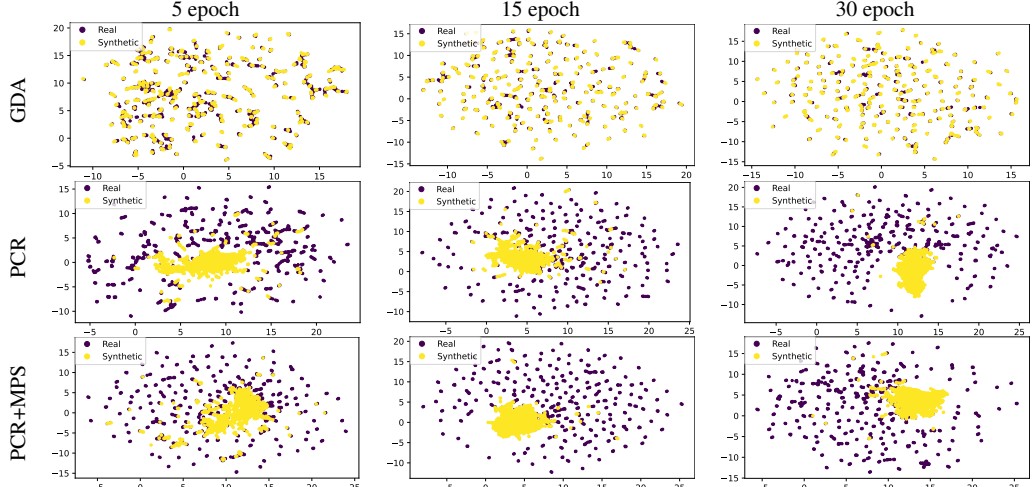

Figure 5: UMAP visualization of feature spaces on training. The plots in the figures represent real and synthetic samples in the feature spaces. Our methods (PCR and PCR+MPS) can help the feature extractors separate real sample clusters. In contrast, the existing method (GDA) confuses the feature extractor by leaking synthetic samples out of the clusters.

## 4.4 Visualization of Feature Spaces

In this section, we discuss the effects of PCR and MPS through the visualizations of feature spaces in training. To visualize the output of $g_\psi$ in 2D maps, we utilized UMAP [33] to reduce the dimensions. UMAP is a visualization method based on the structure of distances between samples, and the low dimensional embeddings can preserve the distance between samples of the high dimensional input. Thus, the distance among the feature clusters on UMAP visualization of $g_\psi$ can represent the difficulty of the separation by $h_\omega$. We used the official implementation by [33][5] and its default hyperparameters. We plotted the UMAP embeddings of $g_\psi(x)$ and $g_\psi(x_{\mathrm{p}})$ at $\{5, 15, 30\}$ epochs, as shown in Figure 5; we used ResNet-18 trained on the Cars dataset. At first glance, we observe that GDA and PCR formed completely different feature spaces. GDA forms the feature spaces by forcing to treat the synthetic samples the same as the real samples through cross-entropy loss and trying to separate the clusters of samples according to the class labels. However, the synthetic samples leaked to the inter-cluster region at every epoch because they could not represent class categories perfectly as discussed in Sec. 1. This means that the feature extractor might be distorted to produce features that confuse the classifier. On the other hand, the synthetic samples in PCR progressively formed a cluster at the center, and the outer clusters can be seen well separated. Since UMAP can preserve the distances between clusters, we can say that the sparse clusters that exist far from the center are considered easy to classify, while the dense clusters close to the center are considered difficult to classify. In this perspective, PCR helps $g_\psi$ to leverage the synthetic samples for learning feature representations interpolating the dense difficult clusters. This is because the synthetic samples tend to be in the middle of clusters due to their less representativeness of class categories. That is, PCR can utilize the partial but useful information contained in the synthetic samples while avoiding the negative effect. Further, we observe that applying MPS accelerates the convergence of the synthetic samples into the center.

For a more straightforward visualization of class-wise features, we designed a simple binary classification task that separates Pets [31] into dogs and cats, and visualized the feature space of a model trained on this task. Fig. 6 shows the feature space after one epoch of training. While GDA failed to separate the clusters for each class, MGR clearly separated the clusters. Looking more closely, MGR helps samples to be dense for each class. This is because PCR makes the feature extractor learn slight differences between the synthetic samples that interpolate the real samples. From these observations, we conclude that our MGR can help models learn useful feature representations for solving tasks.

---

5  https://github.com/lmcinnes/umap

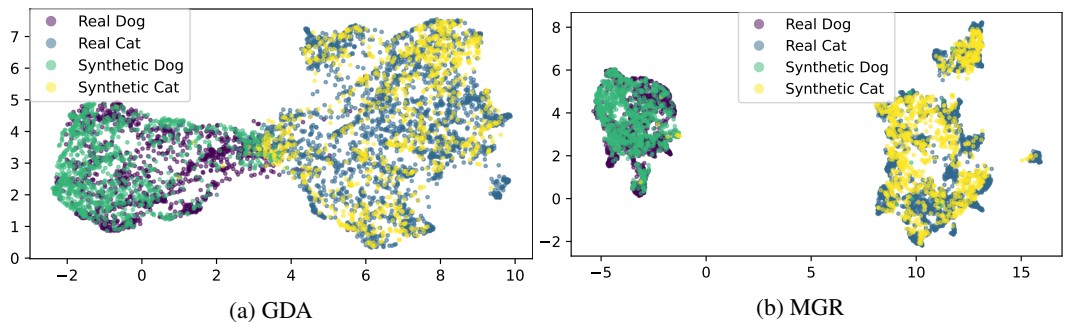

(a) GDA

(b) MGR

Figure 6: UMAP visualization (ResNet-18). We used the Pets dataset by modifying the class definition to the binary classes (dogs and cats). The visualization results are plotted after one epoch training with 2048 real samples and 2048 synthetic samples.

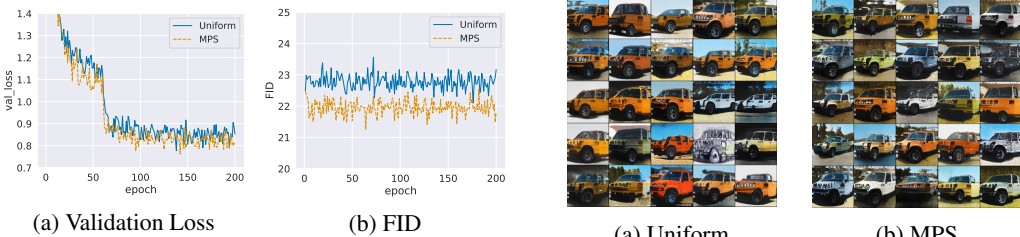

(a) Validation Loss

(b) FID

Figure 7: Statistics in training (Cars)

(a) Uniform

(b) MPS

Figure 8: Synthetic samples (class: `Hummer`)

## 4.5 Analysis of MPS

**Evaluation of validation loss.** We investigate the effects on validation losses when using MPS. Through the meta-optimization by Eq. (7), MPS can generate samples that reduce the validation loss of $f_\theta$. We recorded the validation loss per epoch when applying uniform sampling (Uniform) and when applying MPS. We used the models trained on Cars and applied the PCR loss on both models. Figure 7a plots the averaged validation losses. MPS reduced the validation loss. In particular, MPS was more effective in early training epochs. This is related to accelerations of converging the central cluster of synthetic samples discussed in Section 4.4 and Figure 5. That is, MPS can produce effective samples for regularizing features and thus speed up the entire training of $f_\theta$.

**Quantitative evaluation of synthetic samples.** We evaluate the synthetic samples generated by MPS. To assess the characteristics of the samples, we measured the difference between the data distribution and distribution of the synthetic samples. We leveraged the Fréchet Inception distance (FID, [34]), which is a measurement of the distribution gap between two datasets using the closed-form computation assuming multivariate normal distributions:

$$\text{FID}(\mathcal{D}, \mathcal{D}_\text{p}) = \|\mu - \mu_\text{p}\|_2^2 + \text{Tr}\left(\Sigma + \Sigma_\text{p} - 2\sqrt{\Sigma\Sigma_\text{p}}\right),$$

where $\mu$ and $\Sigma$ are the mean and covariance of the feature vectors on InceptionNet for input $\{x^i\}$. Since FID is a distance, the lower $\text{FID}(\mathcal{D}, \mathcal{D}_\text{p})$ means that $\mathcal{D}_\text{p}$ contains more high-quality samples in terms of realness. We computed FID scores using 2048 samples in $\mathcal{D}$ and 2048 synthetic samples every epoch; the other settings are given in Section 4.1. The FID scores in training are plotted in Figure 7b. We confirm that MPS consistently produced higher-quality samples than Uniform. This indicates that the sample quality is important for generalizing $f_\theta$ even in PCR, and uniform sampling can miss higher quality samples in generative models. Since the performance gain by GDA+MPS in Table 1a did not better than MGR, the higher-quality samples by MPS can still contain uninformative samples for the cross-entropy loss, but they are helpful for PCR learning good feature representations.

**Qualitative evaluation of synthetic samples.** We evaluate the qualitative properties of the synthetic samples by visualizing samples of a class. The samples generated by Uniform and MPS are shown in Figure 8a and 8b, where the dataset was Cars and the class category was `Hummer`. Compared with

Table 2: Performance of MGR varying $G_\Phi$ (ResNet-18, Cars).

| $G_\Phi$ | Conditional | FID | Top-1 Acc. (%) |
|---|---|---|---|
| None (Base Model) | – | – | $85.50^{\pm.10}$ |
| Real CR | – | – | $86.16^{\pm.02}$ |
| FastGAN [35] | No | 23.1 | $86.30^{\pm.16}$ |
| BigGAN [14] | Yes | 15.6 | $86.86^{\pm.06}$ |
| StyleGAN2-ADA [9] | Yes | 9.5 | $87.22^{\pm.15}$ |
| StyleGAN-XL [36] | Yes | 6.2 | $88.37^{\pm.20}$ |

Table 3: Performance comparison between MGR and data augmentation methods (ResNet-18, Cars, Top-1 Acc. (%)).

| Data Augmentation | Base Model | +MGR |
|---|---|---|
| None | $85.50^{\pm.10}$ | $87.22^{\pm.15}$ |
| MixUp [37] | $86.87^{\pm.30}$ | $87.60^{\pm.46}$ |
| CutMix [38] | $86.13^{\pm.19}$ | $87.80^{\pm.51}$ |
| AugMix [39] | $86.25^{\pm.11}$ | $87.65^{\pm.03}$ |
| RandAugment [21] | $87.47^{\pm.05}$ | $88.67^{\pm.10}$ |
| SnapMix [40] | $87.11^{\pm.20}$ | $88.21^{\pm.13}$ |
| TrivialAugment [25] | $87.83^{\pm.16}$ | $\mathbf{89.10^{\pm.19}}$ |

Uniform, MPS produced samples with more diverse body colors and backgrounds. That is, MPS focuses on the color and background of the car as visual features for solving classifications. In fact, since `Hummer` has various colors of car bodies and can drive on any road with four-wheel drive, these selective generations of MPS are considered reasonable in solving classification tasks.

### 4.6  Effect of Generative Models

Here, we evaluate MGR by varying the generative model $G_\Phi$ for producing synthetic samples. As discussed in Sec. 3.1 and 3.2, MGR can use arbitrary unconditional/conditional generative models as $G_\Phi$ unless it has a latent space. To confirm the effects when changing $G_\Phi$, we tested MGR with FastGAN [35], BigGAN [18], and StyleGAN-XL [36]. Table 2 shows the results on Cars. The unconditional FastGAN achieved similar improvements as conditional cases. However, since unconditional generative models are generally of low quality in FID, they are slightly less effective. For the conditional generative models, we observe that MGR performance improvement increases as the quality of synthetic samples improves. These results suggest the potential for even greater MGR gains in the future as the performance of the generative model improves. We also evaluate MGR with recent diffusion models in Appendix B.3. Meanwhile, to evaluate the value of synthetic samples in the consistency regularization, we compare MGR with the case where we use real samples in Eq. (6) (Real CR). Our method outperformed Real CR. This can be because interpolation by the generators helps the feature extractor to capture the difference between images. Furthermore, our method searches the optimal synthetic samples by using latent vectors of generative models with MPS. In contrast, Real CR cannot search the optimal real samples for CR loss because real data is fixed in the data space.

### 4.7  Combination of MGR and Data Augmentation

To assess the practicality of MGR, we evaluate the comparison and combination of MGR and existing data augmentation methods. Data augmentation (DA) is a method applied to real data and is an independent research field from generative data augmentation. Therefore, MGR can be combined with DA to improve performance further. Table 3 shows the evaluation results when comparing and combining MGR and DA methods; we used MixUp [37], CutMix [38], AugMix [39], RandAugment [21], SnapMix [40], and TrivialAugment [25] as the DA methods. The improvement effect of MGR is comparable to that of DA, and the highest accuracy was achieved by combining the two methods. In particular, MGR was stably superior to sample-mix-based such as MixUp and AugMix. This result also indicates that synthetic samples, which non-linearly interpolate the real samples, elicit better performance than linearly interpolating real samples by the sample-mix-based DA methods. We consider this strength to be an advantage of using generative models.

## 5  Related Work

We briefly review generative data augmentation and training techniques using generative models.

The earliest works of generative data augmentation are [19, 41, 42]. They have demonstrated that simply adding synthetic samples as augmented data for classification tasks can improve performance in few-shot learning, person re-identification, and medical imaging tasks. Tran et al.[20] have

proposed a generative data augmentation method that simultaneously trains GANs and classifiers for optimizing $\theta$ to maximize the posterior $p(\theta|x)$ by an EM algorithm. Although this concept is similar to our MPS in terms of updating both $G_\Phi$ and $f_\theta$, it requires training the specialized neural architectures based on GANs. In contrast, MPS is formalized for arbitrary existing generative models with latent variables, and it requires no restrictions to the training objectives of generative models.

On the analysis of generative data augmentation, Shmelkov et al. [7] have pointed out that leveraging synthetic samples as augmented data degrades the performance in general visual classification tasks. They have hypothesized that the cause of the degradation is the less diversity and fidelity of synthetic samples from generative models. Subsequent research by Yamaguchi et al. [8] have shown that the scores related to the diversity and fidelity of synthetic samples (i.e., SSIM and FID) are correlated to the test accuracies when applying the samples for generative data augmentation in classification tasks. Based on these works, our work reconsiders the training objective and sampling method in generative data augmentation and proposes PCR and MPS.

More recently, He et al. [43] have reported that a text-to-image generative model pre-trained on massive external datasets can achieve high performance on few-shot learning tasks. They also found that the benefit of synthetic data decreases as the amount of real training data increases, which they attributed to a domain gap between real and synthetic samples. In contrast, our method does not depend on any external dataset and successfully improves the accuracy of the classifier even when synthetic data are not representatives of class labels (i.e., having domain gaps).

## 6 Limitation

One of the limitations of our method is the requirement of bilevel optimization of classifier and finder networks. This optimization is computationally expensive particularly when used with generative models that require multiple inference steps, such as diffusion models as discussed in Appendix B.3. We have tried other objective functions not requiring bilevel optimization, but at this time, we have not found an optimization method that outperforms MPS (see Appendix B.4). Nevertheless, since recent studies rapidly and intensively focus on the speedup of diffusion models [44, 45, 46], we can expect that this limitation will be negligible in near the future. Additionally, applying MGR to pre-trained text-to-image diffusion models (e.g., [44]) is also important for future work because they have succeeded in producing effective samples for training downstream task models in a zero-shot manner [43].

## 7 Conclusion

This paper presents a novel method for generative data augmentation called MGR. MGR is composed of two techniques: PCR and MPS. To avoid the degradation of classifiers, PCR utilizes synthetic samples to regularize feature extractors by the simple consistency regularization loss. MPS searches the useful synthetic samples to train the classifier through meta-learning on the validation loss of main tasks. We empirically showed that MGR significantly improves the baselines and brings generative data augmentation up to a practical level. We also observed that the synthetic samples in existing generative data augmentation can distort the decision boundaries on feature spaces, and the PCR loss with synthetic samples dynamically generated by MPS can resolve this issue through visualization. We consider that these findings will help future research in this area.

## Acknowledgements

We thank the members of the Kashima Laboratory and the PRMU community for discussing the initial concepts of this paper and giving useful advice.

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

# Appendix

The following manuscript provides the supplementary materials of the main paper: Regularizing Neural Networks with Meta-Learning Generative Models.

## A    Algorithm of Meta Generative Regularization

---

**Algorithm 1** Meta Generative Regularization

---

**Require:** Training dataset $\mathcal{D}$, validation dataset $\mathcal{D}_{\text{val}}$ main model $f_\theta$, generator $G$, finder $F_\phi$, training batchsize $B$, pseudo batchsize $B_{\text{p}}$, validation batchsize $B_{\text{val}}$, step size $\eta$ and $\xi$, hyperparameter $\lambda$ and $\lambda_{\text{KL}}$
**Ensure:** Trained main model $f_\theta$
1:  **while** not converged **do**
2:     $\{(x^i, y^i)\}_{i=1}^B \sim \mathcal{D}$
3:     $\{z^i\}_{i=1}^{B_{\text{p}}} \sim \mathcal{N}(0, I)$
4:     // Updating $\phi$ for MPS
5:     $\{(x_{\text{val}}^i, y_{\text{val}}^i)\}_{i=1}^{B_{\text{val}}} \sim \mathcal{D}$
6:     $\{x_{\text{p}}^i\}_{i=1}^{B_{\text{p}}} = \{G_\Phi(F_\phi(z^i), y_{\text{p}}^i)\}_{i=1}^{B_{\text{p}}}$
7:     $\theta' \leftarrow \theta - \eta \nabla_\theta (\frac{1}{B}\ell(f_\theta(x^i), y^i) + \frac{\lambda}{B_{\text{p}}}\ell_{\text{PCR}}(x_{\text{p}}^i; \psi))$
8:     $\phi \leftarrow \phi - \xi \nabla_\phi (\frac{1}{B_{\text{val}}}\ell(f_{\theta'}(x_{\text{val}}), y_{\text{val}}) + \lambda_{\text{KL}}(D_{\text{KL}}(p_\phi(z) \| p(z))))$
9:     // Updating $\theta$ with PCR
10:    $\{x_{\text{p}}^i\}_{i=1}^{B_{\text{p}}} = \{G_\Phi(F_\phi(z^i), y_{\text{p}}^i)\}_{i=1}^{B_{\text{p}}}$
11:    $\theta \leftarrow \theta - \eta \nabla_\theta (\frac{1}{B}\ell(f_\theta(x^i), y^i) + \frac{\lambda}{B_{\text{p}}}\ell_{\text{PCR}}(x_{\text{p}}^i; \psi))$
12: **end while**

---

## B    Additional Experiments

### B.1    Evaluation of Gradient Approximation

Here, we evaluate the gradient approximation by Eq. (9). As shown in Table 4, Eq. (9) well approximated the second-order gradients in speeding up over 10% with 0.08 of the accuracy drop.

Table 4: Performance comparison between MPS with 2nd-order gradients and 1st-order approximated gradients (ResNet-18, Cars).

| Method | Top-1 Acc. (%) | Wall Clock Time (hours) |
|---|---|---|
| 2nd-Order | $87.30^{\pm.39}$ | 6.55 |
| 1st-Order Approx. | $87.22^{\pm.15}$ | 5.79 |

### B.2    Ablation study of $F_\phi$

In Section 3.2, we introduce $F_\phi$ for meta-optimized parameters and the residual architectures with MLP defined by Eq. (10). We performed an ablation study of MPS with respect to the meta-optimized parameters and the architectures of $F_\phi$. We compared MPS with a variant of MPS optimizing $G_\Phi$ instead of $F_\phi$. We also attempted other architectures for $F_\phi$ including **Linear**: $W_\phi(z) + b$, **MLP**: $\text{MLP}_\phi(z)$, and **Residual+Shallow**: $z + \tanh(W_\phi(z) + b)$. The results of these variations are shown in Table 5. We observed that MPS with $G_\Phi$ caused failures of training $f_\theta$ and degraded the accuracy. On the other hand, all variants of MPS with $F_\phi$ succeeded in boosting the models without MPS. Thus, restricting the number of optimized parameters is important, and determining an optimal $z$ with the finder $F_\phi$ is effective on the optimization problems of MPS. For the variants of MPS with $F_\phi$, we observed that the residual architectures and regularization by $D_{\text{KL}}(p_\phi(z)\|p(z))$ contributed to the successes. Interestingly, MPS with Linear $F_\phi$ outperformed MPS with MLP $F_\phi$, i.e., significantly transforming the input $z \sim p(z)$ by complex functions results in low accuracy. These results suggest that better latent vectors in $\mathcal{Z}$ to train $f_\theta$ can exist near the uniformly sampled input $z$. Thus, limiting the search range by $\tanh$ in the residual architectures can help in finding better latent vectors.

Table 5: Ablation study of MPS (ResNet-18, Cars).1

| Method | Top-1 Acc. (%) |
|---|---|
| Without MPS (PCR) | $86.32^{\pm.07}$ |
| MPS | $\mathbf{87.22^{\pm.15}}$ |
| MPS with $G_\Phi$ | $84.47^{\pm.05}$ |
| MPS with Linear $F_\phi$ | $86.51^{\pm.09}$ |
| MPS with MLP $F_\phi$ | $86.35^{\pm.13}$ |
| MPS with Residual+Shallow $F_\phi$ | $86.88^{\pm.16}$ |
| MPS w/o $D_{\mathrm{KL}}(p_\phi(z)\|p(z))$ | $86.92^{\pm.22}$ |

## B.3 MGR with Diffusion Models

We tested our method on EDM [45], a recent diffusion model. Due to the computation cost, we used a 10% reduced CIFAR-10 as the dataset. We optimized $F_\phi$ to search the first step noise of the diffusion process. Table B-4 shows that our method with EDM improves Base Model. However, the overhead of incorporating diffusion models was significant; it takes more than ten times longer training than GANs. In future work, we will investigate lighter-weight methods using the diffusion model.

Table 6: Performance studies on Diffusion Model (ResNet-18 on Cars)

| Method | Top-1 Acc. (%) |
|---|---|
| Base Model | $86.49^{\pm.48}$ |
| GDA (EDM) | $85.80^{\pm.30}$ |
| MGR | $\mathbf{88.49^{\pm.12}}$ |

## B.4 Updating $F_\phi$ without Meta-optimization

MPS consists of meta-learning on validation losses requiring bi-level optimization, which is a relatively heavy computation. One can consider if $F_\phi$ could be trained without meta-optimization. Here, we try alternative methods other than meta-learning to update $F_\phi$. Instead of meta-learning, we used a strategy of choosing hard examples via optimizing $F_\phi$. That is, we optimize $F_\phi$ by maximizing the training cross-entropy (CE) loss and the PCR loss on synthetic samples. Note that, in both cases, we used the PCR loss for synthetic samples when training classifiers. Table A-2 shows the results. Optimizing $F_\phi$ with CE and PCR slightly improved the baselines but significantly underperformed our method (MGR). This result can justify using meta-optimizing $F_\phi$ to generate useful samples for classifiers. Nevertheless, this idea could inspire a sampling method that does not require bi-level optimization in future work.

Table 7: Performance comparison of updating strategies for $F_\phi$ (ResNet-18 on Cars)

| Method | Top-1 Acc. (%) |
|---|---|
| Base Model | $85.50^{\pm.10}$ |
| PCR | $86.36^{\pm.08}$ |
| Optimizing $F_\phi$ w/ CE | $86.52^{\pm.21}$ |
| Optimizing $F_\phi$ w/ PCR | $86.44^{\pm.68}$ |
| MGR | $\mathbf{87.22^{\pm.15}}$ |

## B.5 ImageNet Classification

We evaluated our MGR on ImageNet by randomly initializing ResNet-18 as the classifier and the pre-trained BigGAN as the generator. Table 8 shows that MGR successfully improves top-1 accuracy, while naive generative data augmentation (GDA) does not; this is the same trend as Table 1 (a) of the main paper. This result indicates that our method consistently works on complex and large-scale datasets. We will add the result in multiple trials with standard deviation to the paper.

Table 8: ImageNet Classification (ResNet-18)

| Method | Top-1 Acc. (%) |
|---|---|
| Base Model | 68.10 |
| GDA | 64.74 |
| MGR | **70.55** |

## B.6 Latent Augmentation

Some readers may wonder why not also utilize augmentation in the generative model's latent space instead of data space. We implemented this approach by adding Gaussian noise to the latent vector as $z' = z + s$, where $s \sim \mathcal{N}(0, 10^{-3})$. Then, we compute the consistency regularization between $g(G(z))$ and $g(G(z'))$, instead of $g(G(z))$ and $g(T(G(z)))$. We call this variant LatentAugment. We found that LatentAugment improves the performance of our MGR (Table 9). Interestingly, LatentAugment can improve when it is used solely without image data augmentation $T$, i.e., RandAugment. This indicates that we can obtain meaningful variants by perturbing latent vectors, which is challenging for conventional data augmentation. However, this approach doubles the number of generators' forward computations and thus leads to increased computation time and memory footprint.

Table 9: Classification on Cars (ResNet-18)

| Method | Top-1 Acc. (%) |
|---|---|
| Base Model | $85.50^{\pm.10}$ |
| MGR (LatentAugment) | $86.49^{\pm.33}$ |
| MGR (RandAugment) | $87.22^{\pm.15}$ |
| MGR (RandAugment + LatentAugment) | $\mathbf{87.85^{\pm.53}}$ |

## B.7 Comparison to Existing Sampling Technique

Here, we evaluate MPS by comparing it with an existing sampling technique called multi-modal truncation sampling [47], which was originally proposed for sampling better quality samples for image generation. Table 10 shows the result of combining multi-modal truncation sampling and PCR. Although the FID score of multi-modal truncation sampling certainly outperforms MPS, the gain of the classification accuracy underperforms MPS. This implies that improving sample quality is a necessary condition for performance improvements, not a sufficient condition. Meanwhile, since MPS explicitly searches for samples that minimize the validation loss, it can improve classifiers more directly than incorporating existing sampling methods.

Table 10: Classification on Cars (ResNet-18)

| Method | Running Mean FID | Top-1 Acc. (%) |
|---|---|---|
| PCR | 22.96 | $86.36^{\pm.08}$ |
| Multi-modal Truncation + PCR | **21.12** | $86.51^{\pm.21}$ |
| MGR (MPS + PCR) | 22.08 | $\mathbf{87.22^{\pm.15}}$ |

## B.8 Performance Study when using Pre-trained Generators

We tried to use ImageNet pre-trained BigGAN and confirmed that this does not solve the degradation problem of GDA (Table 11). However, the use of pre-trained models can enhance our method.

## B.9 Evaluation of Robustness toward Natural Corruption

Our method can improve the robustness against natural corruption and MPS does not generate biased samples toward training distribution. We tested the robustness of our method on CIFAR-10-C [48], which is a test set for CIFAR-10 corrupted by various transformations. Table 12 shows the results.

Table 11: Classification on Cars (ResNet-18, ImageNet pre-trained BigGAN). Underlined scores outperform that of Base Model, and **Bolded scores** are the best among the methods.

| Method / Dataset size | 10% | 25% | 50% | 100% |
|---|---|---|---|---|
| Base Model | $20.11^{\pm.03}$ | $49.33^{\pm.54}$ | $72.91^{\pm.38}$ | $85.80^{\pm.18}$ |
| GDA | $18.82^{\pm.22}$ | $46.38^{\pm.59}$ | $70.23^{\pm.66}$ | $86.11^{\pm.16}$ |
| MGR | $\underline{\mathbf{24.55}}^{\pm.24}$ | $\underline{\mathbf{53.41}}^{\pm.89}$ | $\underline{\mathbf{75.46}}^{\pm.12}$ | $\underline{\mathbf{87.17}}^{\pm.18}$ |

While GDA degraded the performance for all corruptions, PCR significantly improved the base model. Furthermore, MGR achieved even higher robustness. This indicates MPS in MGR can provide samples that are useful for generalization through meta-optimization.

Table 12: Classification on CIFAR-10-C (ResNet-18).

| Method | clean | gaussian noise | shot noise | impulse noise | defocus blur | glass blur | motion blur | zoom blur | snow | frost | fog | brightness | contrast | elastic transform | pixelate | jpeg compression | mean |
|---|---|---|---|---|---|---|---|---|---|---|---|---|---|---|---|---|---|
| Base Model | 86.49 | 53.32 | 53.12 | 39.01 | 34.06 | 37.15 | 31.08 | 36.14 | 46.68 | 36.38 | 19.14 | 52.45 | 10.51 | 41.14 | 46.59 | 53.02 | 42.27 |
| GDA | 84.11 | 52.70 | 52.98 | 39.94 | 29.39 | 38.66 | 28.97 | 28.82 | 45.04 | 40.08 | 24.00 | 50.29 | 13.07 | 40.91 | 45.11 | 52.85 | 41.68 |
| PCR | 87.06 | 59.35 | 60.65 | 37.71 | 47.51 | 50.89 | 41.71 | 47.11 | 63.94 | 57.33 | 31.13 | **73.43** | 13.66 | 58.00 | 63.77 | 68.55 | 53.86 |
| MGR | **88.02** | **61.69** | **62.53** | **41.62** | **52.21** | **51.91** | **47.94** | **51.09** | **65.78** | **59.50** | **34.49** | 73.23 | **14.83** | **60.65** | **65.32** | **71.10** | **56.37** |

