# OpenReview forum: "Regularizing Neural Networks with Meta-Learning Generative Models"
_NeurIPS.cc/2023/Conference — NeurIPS 2023 poster_

### Official Review · Reviewer_Ja7o · 2023-06-27

**Soundness:** 3 good
**Presentation:** 3 good
**Contribution:** 2 fair
**Rating:** 3
**Confidence:** 5

**Summary:**

This paper proposed a regularization method 'Meta generative regularization' based on the bi-level optimization frame addressed for the generative data augmentation. The MGR is consisited of two terms: pseudo consistency regularization (PCR) and meta pseudo sampling (MPS). The training using MGR is formalized  as alternating optimization of a main classification model and finder network for searching latent vectors of generative model(eg, StyleGAN).To maximize the gain from synthetic samples, MGR regularizes a feature extractor part of the classification model using PCR by effectively sampling useful samples for the generalization from GAN using MPS.

**Strengths:**

- Use pseudo consistency regularization to address the distortion of decision boundary.
- Introduce a subnetwork called a finder to improve the training of classifier, and address the unstable training of generator.



**Weaknesses:**

-  The data-driven data augmentation is not novel for the community, e.g., AutoAugment [1], Population Based Augmentation [2], Fast AutoAugment [3], ect. The proposed method is expensive in computation, and only achieve comparable or even worse performance than
hand-designed data augmentation methods, e.g., SnapMix [4]. The advantage of proposed method is not clear to readers.
- The used meta-learning technique is similar to Generative Teaching Networks [5].
- The experimental results can not support the effectiveness of proposed method. The compared method should contain other hand-designed data augmentation methods, e.g., mixup, cutmix, ect; and the data-driven data augmentation methods, e.g., AutoAugment [1], Population Based Augmentation [2], Fast AutoAugment [3], ect. Meanwhile, the hand-designed data augmentation methods SnapMix [4] can achieve a significant improvement on CUB, Cars, Aircraft datasets, while does not introduce additional expensive computation.
- What if the classfication model totally training from scratch on the  synthetic samples generated by the trained finder network for the StyleGAN?
- Compared with existing data-driven data augmentation methods, proposed method is limited in transferability for other classification tasks.

[1] E. D. Cubuk, B. Zoph, D. Mane, V. Vasudevan, and Q. V. Le. AutoAugment: Learning Augmentation Policies from Data. In CVPR, 2018.
[2] D. Ho, E. Liang, I. Stoica, P. Abbeel, and X. Chen. Population Based Augmentation: Efficient Learning of Augmentation Policy Schedules. In ICML, 2019.
[3] S. Lim, I. Kim, T. Kim, C. Kim, and S. Kim. Fast AutoAugment. In NIPS, 2019.
[4] Huang S, Wang X, Tao D. Snapmix: Semantically proportional mixing for augmenting fine-grained data. In AAAI, 2021, 35(2): 1628-1636.
[5] Felipe Petroski Such, Aditya Rawal, Joel Lehman, Kenneth O. Stanley, Jeff Clune. Generative Teaching Networks.  ICML 2020



**Questions:**

See Weaknesses.
- Eq.(9) is wrong, since the numerator does not contain $\epsilon$.

**Limitations:**

The studied problem of this paper is somewhat out-of-date. The effectiveness is limited among existing researches. Especically, the proposed method does not show the effectiveness in some problems with sparse data, e.g., medical imaging.

---

> ### Author Rebuttal · Authors · 2023-08-09
>
> Thank you for your comments on the various points of view.
>
> ### **W1: The data-driven data augmentation is not novel. What is the advantage of the proposed method over existing data augmentation methods?**
> First of all, **the novelty of our work is mainly in solving the performance degradation of generative data augmentation (GDA), not in proposing a new data-driven data augmentation method**. As discussed in Sec. 4.7, data augmentation (DA) and generative data augmentation (GDA) are independent research fields. Thus, they can be combined easily. In this regard, the comparison with DA methods is an indicator of practicality. We have shown that our MGR achieves comparable performance to DA methods and the combination achieves the best performance in Table 3. MGR can outperform the DA baselines by switching the generator to StyleGAN-XL (Table 2), indicating that it continues to improve as generative models evolve in the future. Therefore, **the advantage is a performance improvement that cannot be obtained with DA**. Note that we selected AugMix, RandAugment, and TrivialAugment as the DA baselines because they are more lightweight and powerful baselines than data-driven DA methods such as AutoAugment [f].
>
> [f] Müller, Samuel G., and Frank Hutter. "Trivialaugment: Tuning-free yet state-of-the-art data augmentation." CVPR. 2021.
>
> ---
>
> ### **W2: Difference between the proposed method and Generative Teaching Network (GTN) [g]**
> Thank you for providing related work. **MGR is different and superior to GTN in terms of (I) meta-optimization objective, (II) classifier training, and (III) computation efficiency**. First, MGR meta-optimizes only the finder network for searching optimal samples for classifier training, whereas GTN meta-optimizes entire generators. As a result, MGR avoids the overfitting caused by updating entire generators as reported [h]. Second, MGR trains a classifier with both real and synthetic samples simultaneously, whereas GTN trains it with only synthetic samples. This is also the reason why GTN cannot be a baseline of GDA. This difference comes from the difference in purpose between MGR and GTN: the former is for regularizing classifiers, and the latter is for meta-learning "data generation" for fast adaptation. For the regularization purpose, MGR is a more straightforward method than GTN and GTN could not solve the performance degradation problem of GDA. Third, MGR efficiently computes the objective function through approximating the second-order gradient (Eq. (9)), whereas GTN na\"ively computes meta-gradients. We will add this discussion to related work.
>
> [g] Such, Felipe Petroski, et al. Generative teaching networks: Accelerating neural architecture search by learning to generate synthetic training data. ICML. 2020.
>
> [h] Tero Karras, et al. Training generative adversarial networks with limited data. NeurIPS. 2020
>
> ---
>
> ### **W3: The experimental results can not support the effectiveness of proposed method. The compared method should contain other hand-designed DA methods, e.g., Mixup, CutMix, SnapMix.**
> We would respectfully point out that **we have shown the effectiveness in comparison with DA methods in Sec. 4.7 and Table 3**. Additionally, we provide the results with Mixup and CutMix on Table R-3, indicating **our MGR outperformed Mixup, CutMix, and SnapMix**. Please see also the general response for more details. We will add the results to the paper.
>
> ---
>
> ### **W4: What if the classification model totally training from scratch?**
> Thank you for the comment. **MGR can stably perform even when training classifiers from scratch**. Table R-10 shows the same trend as Table 1. We will add the results to the paper.
>
> **Table R-10. Classification on Cars (Scratch ResNet-18)**
> ||Top-1 Acc.|
> |:-|:-|
> |Base Model|64.29$\pm$.40|
> |GDA|62.93$\pm$.82|
> |MGR|70.62$\pm$.43|
>
> ---
>
> ### **W5: The proposed method is limited in transferability for other classification tasks**
> **We have confirmed that MGR can stably perform on multiple classification datasets in Sec. 4.2 and Table 1 (a)**. Therefore, we consider that MGR has task-wide transferability. If we misunderstand something, we would be happy if you could provide additional comments.
>
> ---
>
> ### **Q1: Eq. (9) is wrong since the numerator does not contain $\epsilon$**
> We appreciate you for pointing out this. In L138, $\theta^{\pm}$ was incorrectly defined and caused confusion. The correct definition is $\theta^{\pm} = \theta \pm \epsilon\eta\nabla_\theta\mathcal{L}_\mathrm{val}(\theta)$. With this modification, $\epsilon$ appears in the numerator, and Eq.(9) becomes the correct definition.
>
> ---
>
> ### **L1: The studied problem of this paper is somewhat out-of-dated**
> We respectfully disagree with this opinion. As the research workshop was held in NeurIPS 2022 [i], machine learning with synthetic samples is an on-going active research topic, not out-of-dated. We believe that our contribution provides new options for the use of synthetic samples and will significantly help develop this research trend.
>
> [i] SyntheticData4ML Workshop. NeurIPS 2022.
>
> ---
>
> ### **L2: The proposed method does not show the effectiveness in medical imaging.**
> We are afraid that it is not the case. We evaluated our method on the Chaoyang dataset [j], which is a medical imaging dataset for classifying cancers. Table R-11 shows the result. We confirm that **our MGR effectively performs on medical imaging dataset**.
>
> **Table R-11. Classification on Chaoyang (ResNet-18, StyleGAN2-ADA)**
> || Top-1 Acc.|
> |:- | :- |
> |Base Model| 83.56$\pm$.57|
> |GDA|84.23$\pm$.57|
> |MGR|**87.48**$\pm$**.15**|
>
> [j] Zhu, Chuang, et al. Hard Sample Aware Noise Robust Learning for Histopathology Image Classification. IEEE transactions on medical imaging.

---

> > ### Comment · Reviewer_Ja7o · 2023-08-12
> > **The concerns are not perfectly addressed**
> >
> > Thanks for your detailed response, most of them has addressed the concerns. However, the major concern is that proposed method need to employ real samples and synthetic samples to train the classifier. However, existing data augmentation methods are easy and lightweight to improve the training. While the proposed method is expensive and cumbersome. Compared with data-driven data augmentation methods, proposed method can only be used for studied task at hand, and can not be transferred to new tasks. In the transfer stage, data-driven data augmentation methods is efficient and lightweight.  Another concern is the proposed method use the data-augmentaion  consistency regularization trick, which is a strong improvement strategy. It is unfair that compared methods do not use such trick to further improve performance. Besides, for data-scarce tasks, e.g., few/zero-shot learning, generated images have obtained SOTA performance, e.g., [1] etc.
> >
> >
> >
> > [1] He R, Sun S, Yu X, et al. Is synthetic data from generative models ready for image recognition?In ICLR, 2023.

---

> > > ### Author Response · Authors · 2023-08-14
> > > **Response for the remaining concerns**
> > >
> > > Thank you for your timely response and additional explanations of your concerns.
> > >
> > > > Thanks for your detailed response, most of them has addressed the concerns.
> > >
> > > We are pleased that our response addressed most of your concerns.
> > >
> > > > the major concern is that proposed method need to employ real samples and synthetic samples to train the classifier.
> > >
> > > This is one of the limitations not only of our method but of all GDA-like approaches, which utilize synthetic samples as additional data. **We believe that our method is worth the additional cost; it consistently improves the baseline models trained on only real samples**, as shown in all experiments of the paper and rebuttal (e.g., Table R-2). We would be very happy if you kindly find this positive aspect of our work.
> > >
> > > > While the proposed method is expensive and cumbersome. Compared with data-driven data augmentation methods, proposed method can only be used for studied task at hand, and can not be transferred to new tasks. In the transfer stage, data-driven data augmentation methods is efficient and lightweight.
> > >
> > > Thank you for the additional explanations. As described in L115-119 of Sec. 3.2, the concept of MPS is to dynamically find useful samples for training classifiers that predict $p(y|x)$. This is intended to achieve task-specific generation and not to generalize across tasks. Even so, the question you implied (does the pre-trained finder transfer between tasks?) is interesting as it could be a hint to improve the efficiency of our method. To investigate this, we evaluated the transferability of a finder pre-trained on ImageNet. Table R-12 shows the result on Cars when we utilize the pre-trained finder on ImageNet without meta-optimization on the task. Surprisingly, the fixed ImageNet pre-trained finder improved PCR models. **Although the best performance was achieved by MGR (PCR + MPS), this indicates that pre-trained finders have the potential of transferability and computation efficiency**. This can be because the sampling strategy learned by the finder in ImageNet (including car images) is partially useful in Cars. In future work, we will try to skip the meta-optimization in this direction and reduce the computation costs. Thank you again for this constructive comment.
> > >
> > > **Table R-12. Classification on Cars (ResNet-18)**
> > > || Top-1 Test Acc.|
> > > |:-|:-|
> > > |Base Model|85.80$\pm$.10|
> > > |PCR|86.36$\pm$.08|
> > > |PCR + Pre-trained Finder|86.97$\pm$.03|
> > > |MGR|**87.22**$\pm$**.15**|
> > >
> > > > Another concern is the proposed method use the data-augmentaion consistency regularization trick, which is a strong improvement strategy. It is unfair that compared methods do not use such trick to further improve performance.
> > >
> > > Thank you for the additional concern. Since the consistency regularization (CR) method on synthetic samples (i.e., PCR) is a part of our proposed method, the comparison with other methods that do not use such CR techniques originally is fair. Nevertheless, confirming the effect of CR on real samples is important to see the difference between synthetic and real samples. Table R-13 shows the results. **Our method was superior to CR on real samples, which means that the meta-optimized synthetic samples are preferred to regularize the classifiers**. We will add these results to the paper.
> > >
> > > **Table R-13. Classification on Cars (ResNet-18)**
> > > || Top-1 Test Acc.|
> > > |:-|:-|
> > > |Base Model| 85.80$\pm$.10|
> > > |CR on real data|86.16$\pm$.02|
> > > |SnapMix|87.11$\pm$.20|
> > > |SnapMix + CR on real data|88.16$\pm$.03|
> > > |**MGR (StyleGAN-XL)**|**88.37**$\pm$**.20**|
> > > |**SnapMix + MGR (StyleGAN-XL)**|**90.15**$\pm$**.07**|
> > >
> > > > Besides, for data-scarce tasks, e.g., few/zero-shot learning, generated images have obtained SOTA performance, e.g., [1] etc.
> > >
> > > The method of [1] utilizes pre-trained text-image generative models (e.g., Stable Diffusion), indicating it is a transfer learning method using synthetic samples. Recent works [k,l] also shows a transfer learning method utilizing both real and synthetic samples from Stable Diffusion in the GDA fashion. In contrast to these methods, our method is a regularization for data-scarce tasks that is independent of neither source datasets nor pre-trained generative models. We would respectfully point out that our work has a different research question from these works: solving the performance degradation problem of generative data augmentation. Thus, our contribution is fundamental for leveraging synthetic samples and can be helpful for enhancing the transfer learning methods in future work.
> > >
> > > [k] Dunlap, Lisa, et al. "Diversify your vision datasets with automatic diffusion-based augmentation." arXiv preprint arXiv:2305.16289 (2023).
> > >
> > > [l] Burg, Max F., et al. "A data augmentation perspective on diffusion models and retrieval." arXiv preprint arXiv:2304.10253 (2023).

---

### Official Review · Reviewer_gssr · 2023-07-05

**Soundness:** 3 good
**Presentation:** 3 good
**Contribution:** 3 good
**Rating:** 5
**Confidence:** 4

**Summary:**

The paper proposed meta-generative regularization (MGR) for improving generative data augmentation. MGR is optimized by alternative training between the main and finder network. To train the main network, contrastive learning is used. To train the finder network, the authors propose the bilevel optimization and approximate the solution.

**Strengths:**

1.	The proposed method does not update the generative model, but rather updates the finder network. Thus, the foundation generative model can be used.
2.	I think that the proposed method does not depend on the type of generative models such as normalizing flow, auto-regressive, gan, and score-based generative models.
3.	FDM to reduce the computation complexity.
4.	The effectiveness of the reduced datasets

**Weaknesses:**

1. CutMix [A] and MixUp [B] rather than AugMix are the strong baselines. These methods are simple, easily implemented, no training, and effective. A comparison is needed.
2. In Sec. 4.4, I am not sure about the interpretation of UMAP. From the visualization, the diversity of the proposed method is reduced. Thus, the loss term of contrastive learning, L_PCR, might be important. How much the performance is improved when the main model is trained on cross entropy and contrastive loss using only real data?
3. The result of diffusion model is in the supplementary. However, the experiment of recent generation models, including text-to-image generator, makes the paper more convincing because it shows promising results [C, D, E, F].
4. How does this method affect the evaluation in terms of robustness, generalizability, or bias? These is possibility to generate the biased samples.

[A] CutMix: Regularization Strategy to Train Strong Classifiers with Localizable Features, ICCV 2019

[B] mixup: Beyond Empirical Risk Minimization, ICLR 2018

[C] IS SYNTHETIC DATA FROM GENERATIVE MODELS READY FOR IMAGE RECOGNITION?, ICLR 2023

[D] Fake it till you make it: Learning transferable representations from synthetic ImageNet clones, 2023

[E] Synthetic Data from Diffusion Models Improves ImageNet Classification, 2023

[F] Generative models improve fairness of medical classifiers under distribution shifts, 2023

**Questions:**

Please refer to the weaknesses.

===

I update my rate from 4 to 5 because my concerns are addressed and the authors will reflect the discussion below to the paper.

**Limitations:**

The limitation is described in Sec. 6 in the main paper.

---

> ### Author Rebuttal · Authors · 2023-08-09
>
> We appreciate your constructive comments and suggestions.
>
> ### **W1: Comparing MGR with CutMix and MixUp**
> Thank you for this suggestion. We provide the comparison in Table R-3. **MGR outperformed CutMix and Mixup**. As well as the cases of other DA methods, **the combination of MGR and CutMix/Mixup produces further improvements**. We will add these results in Table 3 of the paper. We would appreciate it if you take a look at the general responses for more details.
>
> ---
>
> ### **W2: Interpretation of UMAP visualization and the performance of consistency regularization loss with real data**
> > In Sec. 4.4, I am not sure about the interpretation of UMAP. From the visualization, the diversity of the proposed method is reduced.
>
> Thank you for the comments. For a more straightforward interpretation, we additionally tried the UMAP visualization on a simple binary classification task (Fig. I in the attached PDF). MGR clearly separates the clusters for each class and forms each cluster to be dense, indicating that **MGR reduces the distance between samples in the same class rather than reduces the diversity in the entire feature space**.
>
> > the loss term of contrastive learning, L_PCR, might be important. How much the performance is improved when the main model is trained on cross entropy and contrastive loss using only real data?
>
> We agree with your comment: $\mathcal{L}_\text{PCR}$ plays an important role in regularizing classifiers. We compared our methods and the consistency regularization with real data (CR on real data) in Table R-2. **Our PCR and MGR outperformed CR on real data**. This implies that synthetic samples help train classifiers with useful features that are not in the real data. Further, we would emphasize that MPS also contributes to performance improvements by meta-learning, which could not be achieved by CR with real samples. Please also see the general response for more details.
>
> ---
>
> ### **W3: On leveraging pre-trained text-to-image generator**
> > the experiment of recent generation models, including text-to-image generator, makes the paper more convincing because it shows promising results [C, D, E, F].
>
> Thank you for the comments. Indeed, the studies of training classifiers with text-to-image generators are increasing and have become popular in the research community. However, we would respectfully point out that our work has a different research question from these works: solving the performance degradation problem of generative data augmentation. The text-to-image generators such as Stable Diffusion used in [C] are pre-trained on large-scale multi-modal datasets. This means that using these models contains the effect of transfer learning; transfer learning across datasets is not included in our claims. Thus, to separate the effects of our method and transfer learning, we focused on the generative models trained on target data only. Since our results show that the synthetic samples are useful even when the target dataset is only available, our contribution is fundamental for leveraging synthetic samples. In future work, we will develop sampling methods taking transfer learning and text-prompting into account. We will add the above discussions in the Limitation section.
>
> On a related note, the performance study when using fine-tuned generators is discussed in the response for Reviewer g8gt (Q1). Please take a look if you are interested.
>
> ---
>
> ### **W4: How does MGR affect to robustness, generalizability, or bias? These is possibility to generate the biased samples.**
> Thank you for your interesting comments. In short, **our method can improve the robustness against natural corruption and MPS does not generate biased samples toward training distribution.** We tested the robustness of our method on CIFAR-10-C [e], which is a test set for CIFAR-10 corrupted by various transformations. Table R-9 shows the results. While GDA degraded the performance for all corruptions, PCR significantly improved the base model. Furthermore, MGR achieved even higher robustness. This indicates MPS in MGR can provide samples that are useful for generalization through meta-optimization. We will add this result to the paper.
>
> **Table R-9. Classification on CIFAR-10-C (severity: 5)**
> |            | clean | gaussian noise | shot noise | impulse noise | defocus blur | glass blur | motion blur | zoom blur | snow  | frost | fog   | brightness | contrast | elastic transform | pixelate | jpeg compression | mean  |
> | :--------- | :---- | :------------- | :--------- | :------------ | :----------- | :--------- | :---------- | :-------- | :---- | :---- | :---- | :--------- | :------- | :---------------- | :------- | :--------------- | :---- |
> | Base Model | 86.49 | 53.32          | 53.12      | 39.01         | 34.06        | 37.15      | 31.08       | 36.14     | 46.68 | 36.38 | 19.14 | 52.45      | 10.51    | 41.14             | 46.59    | 53.02            | 42.27 |
> | GDA        | 84.11 | 52.70          | 52.98      | 39.94         | 29.39        | 38.66      | 28.97       | 28.82     | 45.04 | 40.08 | 24.00 | 50.29      | 13.07    | 40.91             | 45.11    | 52.85            | 41.68 |
> | PCR        | 87.06 | 59.35          | 60.65      | 37.71         | 47.51        | 50.89      | 41.71       | 47.11     | 63.94 | 57.33 | 31.13 | **73.43**      | 13.66    | 58.00             | 63.77    | 68.55            | 53.86 |
> | **MGR**        | **88.02** | **61.69**          | **62.53**      | **41.62**        | **52.21**        | **51.91**      | **47.94**       | **51.09**     | **65.78** | **59.50** | **34.49** | 73.23      | **14.83**    | **60.65**             | **65.32**    | **71.10**            | **56.37** |
>
> [e] Hendrycks, Dan, and Thomas Dietterich. "Benchmarking neural network robustness to common corruptions and perturbations." International Conference on Learning Representations (2019).

---

> > ### Comment · Area_Chair_WAcW · 2023-08-18
> > **Reviewer gssr**
> >
> > Dear Reviewer,
> >
> > The author has posted their rebuttal, but you have not yet posted your response. Please post your thoughts after reading the rebuttal and other reviews as soon as possible. All reviewers are requested to post this after-rebuttal-response.

---

> > > ### Comment · Reviewer_gssr · 2023-08-18
> > > **Response of Rebuttal**
> > >
> > > Thank the authors for the comprehensive rebuttal. My concerns are addressed, and I read other responses. I suggest that the authors move the visualization of UMAP to supplementary materials and reflect on other conducted experiments in the main paper.
> > >
> > > Although the paper [A] was not published before submitting this paper, can the authors discuss [A]? This paper transforms the input space of the function (Generative model) for optimization efficiency. I think that it can validate the mechanism of Finder in this paper and make the introduction of Finder reasonable.
> > >
> > > [A] Landscape Learning for Neural Network Inversion, ICCV, 2023

---

> > > > ### Author Response · Authors · 2023-08-18
> > > >
> > > > Thank you for reading our rebuttal! We are pleased to know that the responses address your concerns.
> > > >
> > > > > I suggest that the authors move the visualization of UMAP to supplementary materials and reflect on other conducted experiments in the main paper.
> > > >
> > > > As you suggested, we will revise the paper with additional experiments. Thank you again!
> > > >
> > > > > Although the paper [A] was not published before submitting this paper, can the authors discuss [A]? This paper transforms the input space of the function (Generative model) for optimization efficiency. I think that it can validate the mechanism of Finder in this paper and make the introduction of Finder reasonable.
> > > >
> > > > Thank you for providing important related work. Indeed, the method of [A] is helpful for explaining the mechanism of the finder because it shows that transforming latent variables according to loss functions efficiently reaches the desired outputs. We will add this discussion to the paper as an intuitive reason why the finder achieves better latent variables through meta-optimization. We appreciate your constructive suggestion.

---

### Official Review · Reviewer_g8gt · 2023-07-06

**Soundness:** 4 excellent
**Presentation:** 4 excellent
**Contribution:** 3 good
**Rating:** 7
**Confidence:** 3

**Summary:**

The authors propose a method for using synthetic images from GANs to augment training image classifiers. The naive approach to this problem is to generate samples for each class and treat these as supervised examples, but this can degrade performance due to image artifacts. Instead, the authors propose to use the generated data only for consistency regularization of the featurizer (”PCR”). Further, they meta-learn the codes to use for generation (”MPS”). On 6 datasets, the authors see performance improvements from using synthetic data this way, rather than performance drops.

**Strengths:**

- The motivation is clear, the method is clean, and the approach makes intuitive sense. The overall paper presentation is very good.
- The authors select a nice set of baselines and show their method outperforms them. In particular, GDA + MH was an important comparison to have.
- The paper contains a solid set of ablations, including: the importance of PCR v. MPS (either can improve performance, both are best), the effect of MPS on sample quality as measured by FID, experiments with multiple generative models for one dataset, and experiments showing that gains are additive with standard data augmentations.

**Weaknesses:**

- A missing baseline is consistency regularization without generative data augmentation, i.e. equation 5 applied to real examples $\mathcal D$ rather than $\mathcal D_p$. How does the method compare to training with this objective, and are gains additive?
- Standard training with TrivialAugment / RandAugment outperform this method on Cars, presumably with much lower computational cost — though the authors do show these can be combined with the method to obtain further gains. I’d like to check if this trend holds for other datasets as well.
- Because of the metalearning loop, the method is too slow to apply with diffusion models, as noted in the Limitations and Appendix B.3. All experiments are done with GANs trained from scratch on the target datasets.
- Experiments are done on relatively small datasets. It would be interesting to compare this method on ImageNet.


**Questions:**

A point of confusion for me is why experiments avoided using pretrained generative models / finetuning from them. As noted in the related work, massive pretraining may help produce higher quality samples. The authors attribute degradations after training on generated data to *class leakage*, i.e. samples where artifacts of multiple classes are combined into the same image, such as the tennis ball with a dog face generated by BigGAN. This seems like the kind of artifact that improved generative models can remove. Does generative data augmentation with large pretrained models result in such performance drops? Have the authors tried their method with finetuned GANs?

**Limitations:**

The authors note that the meta-learning loop renders the method unusable for computationally intensive generative models, e.g. diffusion models.

---

> ### Author Rebuttal · Authors · 2023-08-09
>
> We sincerely appreciate your carefully reading and thoughtful feedback.
>
> ### **W1: How does the proposed method compare to the method using consistency regularization on real samples?**
> Thank you for your insightful suggestion. Although the consistency regularization with real samples brings some improvement, **the proposed method using synthetic samples and meta-learning outperforms this baseline**. We would emphasize that our method has the advantages of performance improvements by meta-learning of the sampling networks and switching the generative model to a better one, which could not be achieved by CR with real samples. For details, we would be happy if you take a look at the general response and Table R-2.
>
> ---
>
> ### **W2: What if comparing our method with DA on other datasets?**
> Thank you for the reasonable comment. Table R-6 and R-7 shows Aircraft and Birds; DA and our MGR were competitive and the combination achieved further accuracy improvements. Thus, **our method provides practical performance regardless of the datasets**. MGR outperformed the DA methods solely for some datasets and consistently achieved the best accuracy when it is combined with DA. We will add these results to the paper.
>
> **Table R-6. Classification on Aircraft (ResNet-18)**
> | |No TDA|AugMix|RandAug|Trivial Aug|
> | :- | :- | :- | :- | :- |
> |Base Model|62.61$\pm$.79|64.53$\pm$.70|63.12$\pm$.52|66.14$\pm$.24|
> |MGR|65.11$\pm$.57|65.65$\pm$.12|64.98$\pm$.09|68.17$\pm$.34|
>
> **Table R-7. Classification on Birds (ResNet-18)**
> || No TDA          | AugMix          | RandAug         | Trivial Aug     |
> |:-|:-|:-|:-|:-|
> |Base Model|72.24$\pm$.32|72.44$\pm$.11|70.80$\pm$.37|73.61$\pm$.19|
> |MGR|74.24$\pm$.34|74.92$\pm$.62|71.14$\pm$.23|75.02$\pm$.29|
>
> ---
>
> ### **W3: Limitations of the computational cost of meta-learning**
> We thank you for carefully reading Limitation and Appendix B.3. As we mentioned in Limitation, diffusion models are rapidly speeding up by recent intensive efforts. We can easily imagine that diffusion models achieve a sampling speed comparable to GANs in the near future because the speed-up issue is also important in other applications, such as real-time rendering. For example, a recent work [d] proposed a diffusion-like generative model called consistency model, which achieves high-quality samples (6.2 of FID in ImageNet) in a few steps. In this perspective, the limitation of training speed on diffusion models will be resolved in the near future.
>
> [d] Song, Yang, et al. "Consistency Models." International conference on machine learning (2023).
>
> ---
>
> ### **W4: It would be interesting to compare this method on ImageNet**
> Thank you for the suggestion. We provide the ImageNet results in Table R-1 in the general response. We confirm that **our method stably performs even on the large-scale dataset**. We would be happy if you could see the general response for more details.
>
> ---
>
> ### **Q1: Does naive generative data augmentation with large pretrained models result in such performance drops? Have the authors tried their method with finetuned GANs?**
> **Yes and Yes. We have tried to use ImageNet pre-trained BigGAN and confirmed that this does not solve the degradation problem of GDA (Table R-8). However, the use of pre-trained models can enhance our method.** We omitted this result from the paper to separate the proposed method's effects from the transfer learning effects by the pre-trained models. Even so, since it is an important fact that fine-tuning does not solve the problem, we will add this result to Appendix. Thank you for pointing out this.
>
> **Table R-8. Classification on Cars (ResNet-18, ImageNet pre-trained BigGAN)**
> || 10% | 25% | 50% | 100%
> -- | -- | -- | -- | --
> Base Model|20.11$\pm$.03|49.33$\pm$.54|72.91$\pm$.38|85.80$\pm$.18
> GDA|18.82$\pm$.22|46.38$\pm$.59|70.23$\pm$.66|86.11$\pm$.16
> MGR|24.55$\pm$.24|53.41$\pm$.89|75.46$\pm$.12|87.17$\pm$.18

---

> > ### Comment · Reviewer_g8gt · 2023-08-16
> >
> > Thanks to the authors for their response. I believe my concerns have been thoroughly addressed.

---

> > > ### Author Response · Authors · 2023-08-17
> > >
> > > Thank you for reading our rebuttal! We are glad to hear that the response addressed your concerns. Thank you again for your constructive and thoughtful feedback.

---

### Official Review · Reviewer_ZFum · 2023-07-07

**Soundness:** 3 good
**Presentation:** 3 good
**Contribution:** 2 fair
**Rating:** 7
**Confidence:** 5

**Summary:**

This paper leverages synthetic images from generative models to train classifier models, effectively using them as an augmentation tool. However, instead of incorporating these images in a simplistic fashion, the synthetic images are utilized as a regularizer. The paper posits that synthetic samples may not always perfectly mirror class categories found in real data distribution. As a response, a meta-learning framework is employed to dynamically ascertain which synthetic samples should be used to minimize validation losses. The paper introduces a feature-based consistency regularization loss for the generated images, a method similar to self-supervised techniques. The unique contribution of this paper is the proposition of a finder network that is trained within a meta-learning context, this network is designed to select images that will enhance the performance of the classification model. Through extensive experimentation, the paper illustrates that the proposed methodology can effectively circumvent the performance deterioration often linked with naive generative data augmentation while concurrently improving the baselines.

**Strengths:**

* The paper is well-presented and easy to follow.
* The paper presents a novel sampling network that is trained in a meta-learning setting to select synthetic images that will enhance the performance of the classification model.
* Through a series of experiments, the paper successfully illustrates the effectiveness of the overall approach, as well as the contributions of different aspects of the methodology, i.e., consistency regularization loss and Meta Pseudo Sampling.
* The proposed generative model augmentation method complements conventional data augmentation methods and can be employed together to enhance performance further.

**Weaknesses:**

* The paper showcases its experiments predominantly on simple, fine-grained classification datasets. To bolster the credibility of their methodology, the authors would benefit from demonstrating how their approach enhances classification performance on more complex, large-scale datasets, such as ImageNet.
* In Table 1, I suggest including an additional baseline: the base model + SSL. This means applying a consistency regularization loss on the real training images without utilizing any synthetic images. This will highlight the degree of improvement achieved using the generative model and the proposed model. The author acknowledges that training the sampling network in a meta-learning setting is computationally expensive, so this comparison would clarify whether the benefits of this approach justify its cost, particularly if the gains are minimal.
* For the consistency loss, the paper solely employs augmentation in the image space. However, one might ask why not also utilize augmentation in the generative model's latent space by introducing small noise to the latent vector z. Given that the latent space of the generators implicitly represents the image manifold, performing augmentation in this latent space could lead to meaningful augmentation in the image space. Such augmentation could potentially be more challenging to achieve using conventional image augmentation methods.
* Figure 5 could be improved further by incorporating a visualization of different class features along the decision boundary of the classifier. This enhancement would provide further insight into how these synthetic images influence the decision boundary under different cases.
* (Just a suggestion) In lines 45-50, I would like to introduce an additional argument. Despite generative models being trained to model p(x), they frequently fail to capture the entire distribution of the training set, focusing mainly on the high-density region of the distribution. Therefore, the information extracted from these generative models is often less comprehensive than the information found in the actual dataset distribution. Furthermore, these synthetic images frequently exhibit disfigured object shapes; for instance, a majority of human shapes in StyleGAN-XL images appear distorted.

**Questions:**

*The major novelty of the paper resides in the proposition of a sampling network, trained via a meta-learning method, to generate synthetic samples aimed at enhancing the classification model's performance. The authors further highlight that a superior FID score from the generative model correlates to improved performance in classification models (Table 2), and the FID score derived from their sampling network significantly surpasses that of uniform sampling (Fig 6b). In the generative models' literature, there exist works that advocate for effective sampling strategies from pre-trained GAN models, utilizing the multimodal truncation method, such as Makady et al. 2022's 'Self-Distilled StyleGAN: Towards Generation from Internet Photos.' How would their sampling approach fare when combined with the consistency regularization loss? It could significantly bolster the paper's case if authors demonstrated that the proposed sampling network exceeds the performance of other sampling methods, like the multimodal truncation method.

**Limitations:**

The author themselves pointed out that the limitation of the method is the requirement of bilevel optimization of classifier and finder networks. This optimization is computationally expensive and may be less practical when used in generative models that require multiple inference steps, such as diffusion models.

---

> ### Author Rebuttal · Authors · 2023-08-09
>
> We appreciate your careful reading and many insightful comments.
>
> ### **W1: Is the proposed method effective on more complex, large-scale datasets such as ImageNet?**
> **Yes.** By following your suggestion, we evaluated our methods on ImageNet. We confirmed the same trend as Table 1 even on ImageNet, which is a large-scale and complex dataset. Please refer to the general response and Table R-1.
>
> ---
>
> ### **W2: Additional baseline of consistency regularization loss with real data**
> Thank you for this suggestion. We examined this baseline in Table R-2 in the above general response and found that **the consistency regularization (CR) with synthetic samples with meta-learning is more effective than one with real samples**. We would emphasize that our method has the advantages of performance improvements by meta-learning of the sampling networks and switching the generative model to a better one, which could not be achieved by CR with real samples. From these facts, we conclude that the proposed approach is beneficial for the cost.
>
> ---
>
> ### **W3: Why not also utilize augmentation in the generative model's latent space?**
> This is a very interesting idea. We implemented this approach by adding Gaussian noise to the latent vector as $z^\prime = z + s$ where $s\sim\mathcal{N}(0,10^{-3})$. Then, we compute the consistency regularization between $g(G(z))$ and $g(G(z^\prime))$, instead of $g(G(z))$ and $g(T(G(z)))$. We call this variant Latent Augmentation. We found that Latent Augmentation improves the performance of our MGR (Table R-4). Interestingly, Latent Augmentation can improve when it is used solely without image data augmentation $T$ (RandAugment). This indicates that we can obtain meaningful variants by perturbing latent vectors, which is challenging for conventional data augmentation, as you said. However, this approach doubles the number of generators' forward computations and thus leads to increased computation time and memory footprint. We would like to present the latent augmentation technique in the paper as a promising option to improve MGR.
>
>
> **Table R-4. Classification on Cars (ResNet-18)**
> || Top-1 Acc.|
> | :- | :- |
> |Base Model| 85.80 $\pm$ .10 |
> |MGR (Latent Augment)| 86.49 $\pm$ .33 |
> |MGR (RandAugment)| 87.22 $\pm$ .15|
> |MGR (RandAugment + Latent Augment)| 87.85 $\pm$ .53 |
>
> ---
>
> ### **W4: UMAP visualization should be improved (Fig. 5)**
> Thank you for the helpful comments. According to your suggestion for a more straightforward visualization of the decision boundary, we conducted the additional visualization study by using a binary classification dataset created by modifying Pets. In Fig. I of the attached PDF, we can see how the proposed method effectively enlarges the margins of the class boundaries. Please also see the general response. We would be happy to receive any comments on the modified visualization results.
>
> ---
>
> ### **W5: Suggestion of additional explanation for the second hypothesis (L45-50)**
> > Despite generative models being trained to model p(x), they frequently fail to capture the entire distribution of the training set, focusing mainly on the high-density region of the distribution. Therefore, the information extracted from these generative models is often less comprehensive than the information found in the actual dataset distribution. Furthermore, these synthetic images frequently exhibit disfigured object shapes; for instance, a majority of human shapes in StyleGAN-XL images appear distorted.
>
> We appreciate your detailed suggestion with the concrete instance. It makes sense that the na\"ive synthetic samples lack detailed information due to focusing on the high-density regions. In fact, from our sample visualization results in Fig. 7, it appears that uniform sampling produces samples with similar features, not comprehensive. The proposed method seems to reduce the negative effect of such disfigured samples by PCR and remove the disfigured samples by MPS. We would gladly incorporate your suggestions into the paper!
>
> ---
>
> ### **Q1: How would multi-modal truncation sampling [c] fare when combined with PCR?**
> Thank you for the question. Through the additional experiments, we found that multi-modal truncation sampling is somewhat helpful for improving PCR, but MPS is more effective. Table R-5 shows the result of combining multi-modal truncation sampling and PCR. **Although the FID score of multi-modal truncation sampling certainly outperforms MPS, the gain of the classification accuracy underperforms MPS**. This implies that improving sample quality is a necessary condition for performance improvements, not a sufficient condition. Meanwhile, since MPS explicitly searches for samples that minimize the validation loss, it can improve classifiers more directly than incorporating existing sampling methods. We will add this analysis to the paper.
>
> **Table R-5. Classification on Cars (ResNet-18)**
> || Running Mean FID | Top-1 Acc.|
> | :- | :- | :- |
> |PCR|22.96| 86.36$\pm$.08|
> |Multi-modal truncation sampling + PCR|21.12|86.51$\pm$.21|
> |MGR (MPS + PCR)|22.08|87.22$\pm$.15|
>
> [c] Mokady, Ron, et al. "Self-distilled stylegan: Towards generation from internet photos." ACM SIGGRAPH 2022 Conference Proceedings. 2022.

---

> > ### Comment · Reviewer_ZFum · 2023-08-18
> > **Thank you for the detailed rebuttal**
> >
> > My apologies in not responding earlier. Thank you very much for taking the time and answering my questios. I have also gone through the other reviewers comments and I am now willing to change my score and recommend acceptance of the paper.

---

> > > ### Author Response · Authors · 2023-08-18
> > >
> > > Thank you for reading our rebuttal and updating the score! We sincerely appreciate again your constructive and insightful suggestions, such as additional baselines and explanations of our hypothesis.

---

### Author Rebuttal · Authors · 2023-08-09

# General Response
We greatly appreciate the reviewers for providing many constructive and insightful comments. We are happy to find all reviewers give scores of 3 (good) or better for the soundness and presentation of our paper. We are also pleased that most reviewers recognize the effectiveness of our method in the Strength section. The feedback from the reviewers concentrates on the suggestions to add experiments to enhance our claims. Thus, we have conducted as many additional experiments as possible to address the concerns. The evaluations on new datasets (e.g., **ImageNet** and **medical imaging**) and baselines (e.g., **CutMix** and **Mixup**) further clarify the performance superiority of our method besides our original claims. For details, please refer to the individual responses.

In the rest of this response, we provide the parts of additional experiments for answering the shared concerns among the reviewers.

## **Additional experimental results for shared concerns**
---

### **Is the proposed method effective on ImageNet? (Reviewers ZFum and g8gt)**
**Yes, our method (MGR) improves top-1 accuracy on ImageNet.** We evaluated our meta generative regularization (MGR) on ImageNet by randomly initialized ResNet-18 as the classifier and the pre-trained BigGAN as the generator. The experimental setup follows [a]. Table R-1 shows that MGR successfully improves top-1 accuracy, while naive generative data augmentation (GDA) does not; this is the same trend as Table 1 (a) of the main paper. This result indicates that our method consistently works on complex and large-scale datasets. We will add the result in multiple trials with standard deviation to the paper.

**Table R-1. ImageNet Classification (ResNet-18)**
||Top-1 Acc. (ImageNet Val.)|
| :-- | :--|
|Base Model |68.10|
|GDA|64.74|
|MGR (Ours)|**70.55**|

[a] Pytorch - Models and Pre-trained Weights

---
### **What if using real data instead of synthetic samples in consistency regularization? (Reviewers ZFum and g8gt)**
Reviewers **ZFum** and **g8gt** recommended assessing the models' performance with the consistency regularization (CR) computed on *real* datasets. We agree that this baseline is important to evaluate the value of synthetic samples in the regularization. Table R-2 shows the performance of the real consistency regularization (we refer to CR on real data) on Cars dataset; the setting is shared with Table 1 of the main paper. For CR on real data, we used the training dataset for computing Eq. (5). **Our pseudo consistency regularization (PCR) slightly outperformed CR on real data**. This can be because interpolation by the generators helps the feature extractor to capture the difference between images. **Furthermore, PCR enables MGR to search the optimal synthetic samples by using latent vectors of generative models**. In contrast, CR on real data cannot search the optimal real samples for CR loss because real data is fixed in the data space. We will add CR on real data results to Table 1 and the above discussions to the paper.

**Table R-2. Classification on Cars (ResNet-18)**
|| Top-1 Test Acc.|
| :--------- | :-------------- |
| Base Model | 85.80 $\pm$ .10 |
| CR on real data | 86.16 $\pm$ .02 |
| PCR | 86.36 $\pm$ .08 |
| MGR | 87.22 $\pm$ .15 |
| MGR (StyleGAN-XL)| **88.37** $\pm$ **.20**|

---

### **Can the proposed method outperform CutMix, MixUp, and SnapMix? (Reviewers gssr and Ja7o)**
**Yes.** We evaluated CutMix, MixUp, and SnapMix on Cars with ResNet-18 (Table R-3). Our MGR outperformed them. Additionally, MGR boosted the performance of them. This trend is consistent with Table 3. This result also indicates that synthetic samples, which non-linearly interpolate the real samples, elicit better performance than linearly interpolating real samples in the input space by CutMix, Mixup, and SnapMix. We consider this strength is an advantage of using generative models.

**Table R-3. Classification on Cars (ResNet-18)**
||Top-1 Acc.|
|:-|:-|
|Base Model|85.50$\pm$.10|
|CutMix|86.13$\pm$.19|
|Mixup|86.87$\pm$.30|
|SnapMix|87.11$\pm$.20|
|**MGR**|87.22$\pm$.15|
|**MGR+CutMix**|87.80$\pm$.51|
|**MGR+Mixup**|87.60$\pm$.46|
|**MGR+SnapMix**|88.21$\pm$.13|

---

### **Additional UMAP visualization on binary classification dataset (Reviewers ZFum and gssr)**
Reviewers **ZFum** and **gssr** commented on the modification and interpretation of UMAP visualization in Fig. 5. For a more straightforward visualization, we designed a simple binary classification task that separates Pets [b] into dogs and cats, and visualized the feature space of a model trained on this task. Fig. I in the attached PDF shows the feature space after one epoch of training. While GDA failed to separate the clusters for each class, **MGR clearly separates the clusters**. Looking more closely, MGR helps samples to be dense for each class. This is because PCR makes the feature extractor learn slight differences between the synthetics samples that interpolate the real samples.

[b] Parkhi, Omkar M., et al. "Cats and dogs." 2012 IEEE conference on computer vision and pattern recognition. IEEE (2012).

---

---

### Decision · Program_Chairs · 2023-09-21

**Decision:**

Accept (poster)

**Comment:**

The paper received mixed ratings, there was a rebuttal: Four knowledgeable reviewers recommended: Borderline Accept, Accept, Accept, Borderline Accept, and Reject. On balance, it is our recommendation to accept the paper. No basis to overturn the reviews. Authors should attend to the main points in the reviews. when preparing a final version. No basis to overturn the reviews.